# FOVI: A biologically-inspired foveated interface for deep vision models

Nicholas M. Blauch [1] [2]  George A. Alvarez [1]  Talia Konkle [1]

## Abstract

Human vision is foveated, with variable resolution peaking at the center of a large field of view; this reflects an efficient trade-off for active sensing, allowing eye-movements to bring different parts of the world into focus with other parts of the world in context. In contrast, most computer vision systems encode the visual world at a uniform resolution, raising challenges for processing full-field high-resolution images efficiently. We propose a foveated vision interface (FOVI) based on the human retina and primary visual cortex (V1), that reformats a variable-resolution retina-like sensor array into a uniformly dense, V1-like sensor manifold. Receptive fields are defined as k-nearest-neighborhoods (kNNs) on the sensor manifold, enabling kNN-convolution via a novel kernel mapping technique. We demonstrate two use cases: (1) an end-to-end kNN-convolutional architecture, and (2) a foveated adaptation of the DINOv3 ViT foundation model, leveraging low-rank adaptation (LoRA). These models provide competitive performance with a fraction of the pixels and computational cost of full resolution non-foveated baselines, opening pathways for efficient and scalable active sensing for high-resolution egocentric vision. Code[†] and pre-trained models[‡] are available.

## 1. Introduction

Processing the visual world in its native high resolution poses serious computational challenges. Notably, within deep learning, computer vision has classically simplified the problem by working with low resolution images, with 224x224 being typical (Krizhevsky et al., 2012; Deng et al., 2009). However, processing at higher resolution

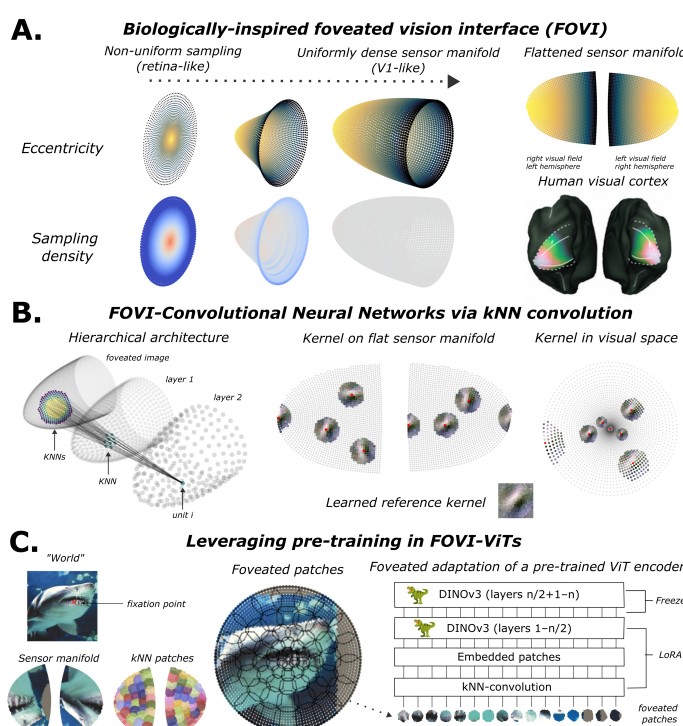

**Figure 1. A.** Foveated sensing as uniform sampling on a magnified sensor manifold (Rovamo & Virsu, 1984). With manifold divided at the vertical meridian and flattened, its relationship to the two hemispheres of primary visual cortex is evident. **B.** Building hierarchical convolutional networks via uniform k-nearest-neighbor (kNN) sampling on the sensor manifold. Kernel mapping allows for kNN-convolution over the sensor manifold, yielding denser and smaller RFs in the fovea. **C.** Building vision transformers (ViTs) from a kNN-convolution-based patchification of the sensor manifold. Low-rank adaptation allows for successful adaptation of off-the-shelf foundation models as efficient foveated variants.

while being sensitive to a large field-of-view is critical to many real-world tasks, including applications in humanoid robotics, self-driving cars, and broader scene processing. For transformer-based architectures employing self-attention (Dosovitskiy et al., 2020), the cost of increasing the resolution of an image (i.e., its side length) imposes doubly quadratic costs; first, in the number of pixels produced, and second, in the attention operation, where each image patch attends to all other patches. These costs remain a barrier to widespread adoption of high-resolution vision

[1]Department of Psychology and Kempner Institute, Harvard University, Cambridge MA [2]NVIDIA. Correspondence to: Nicholas M. Blauch <nblauch@gmail.com>.

*Proceedings of the $43^{rd}$ International Conference on Machine Learning*, Seoul, South Korea. PMLR 306, 2026. Copyright 2026 by the author(s).

[†]https://github.com/nblauch/fovi
[‡]https://huggingface.co/fovi-pytorch

models. Efficient solutions are thus a priority.

Here, we turn to the human visual system for inspiration, as it can process information at very high resolutions near the center of gaze (the fovea), while simultaneously representing a very large field-of-view (˜180°) with progressively lower resolution moving farther from the center of gaze. These changes in acuity are attributed to the high density of both cone photoreceptors and retinal ganglion cells (RGCs) in the fovea, and the progressively lower density of both as distance from the fovea increases (Watson, 2014). Information from the retina is mapped to a uniform representation via the thalamus into the primary visual cortex (V1), where more cortical area is dedicating to foveal vs. peripheral regions (known as "cortical magnification"; (Daniel & Whitteridge, 1961). Thus, not only does the human visual system capture more detailed images at the fovea, but it also devotes more cortical real-estate to processing information at the center of gaze (Schwartz, 1980).

Why is the human visual system foveated? Some back-of-the-envelope calculations provide some intuition (details are provided in Figure S1). If we first assume that V1 were to dedicate as much space to the full visual field as it normally does to central vision, this would result in the long-axis of V1 expanding to 1.2 meters – approximately 25x its typical length – catastrophically increasing both space and energy demands. If we instead assume a fixed amount of cortical real-estate with the peak resolution throughout, only a 3° field-of-view would be possible, impairing broader visual interactions with the world, such as navigation and detection. Thus, foveated sampling of the visual environment provides a balanced solution to the trade-off between resolution, field-of-view, and spatial efficiency, opting for high-resolution over a limited field-of-view, and lower resolutions over larger fields-of-view, with a modest spatial and energetic cost.

**Conflict of Interest Disclosure** N.M.B is employed by NVIDIA, which is developing extensions of FOVI.

## 2. Prior work

While foveated sensing is not a dominant approach in computer vision, it has historically received substantial interest (Weiman & Chaikin, 1979; Javier Traver & Bernardino, 2010; Wang et al., 2021; Da Costa et al., 2024; Jérémie et al., 2024). Generally, two forms of foveated sensing have been explored: a log-polar image model (Weiman & Chaikin, 1979; Javier Traver & Bernardino, 2010; Jérémie et al., 2024), which samples radius (eccentricity) logarithmically and selects an equal number of angular samples at each radius, and warped Cartesian approaches (Basu & Licardie, 1995; Lukanov et al., 2021; Wang et al., 2021; Da Costa et al., 2024), which re-project log-polar images

back into Cartesian space to produce a warped image that over-represents the center of gaze. These approaches share a common issue, derived from their attempt to produce a rectangular grid-like foveated image. The result is locally *anisotropic* sampling (locally, the sampling rate differs with respect to polar angle and radius), a non-biologically-plausible property that produces undesirable warped receptive field shapes (Figures S3, S4). A notable exception is the recent model of Killick et al. (2023); however, their approach requires the use of fixed Gaussian-derivative basis functions in place of learned spatial kernels, and is thus not directly comparable to the other methods which allow for end-to-end learning of both spatial and feature-based representations. Similarly, Cheung et al. (2017) demonstrated the emergence of foveated sampling in a scenario where the sampling grid was learned, but the irregular structure of the learned grid did not support convolutional processing. We discuss these approaches in greater detail in Appendix 10.3. Last, some approaches have foregone a consideration of foveated *sensing*, and have modeled foveated perceptual processing purely at the architectural level, by focusing resources more on central vs. peripheral image content (Kerr et al., 2025; Chuang et al., 2025), showing benefits in robotics applications. While promising, these approaches leverage a discrete set of processing resolutions rather than a continuum, and are not designed with sensing efficiency in mind. A foveated sensor can provide additional efficiency gains, particularly in sim2real pipelines (Pinto & Gupta, 2016), where reducing the numbers of rays traced can reduce both computational and memory-based resources. Overall, prior work has introduced a variety of mechanisms for implementing foveation in computer vision models, but there is not yet a general-purpose implementation that shows well-behaved and biologically-plausible visual field sampling, that can be adopted across diverse architectures.

## 3. Summary of contributions

1. We introduce a new *foveated vision interface* to deep vision models (FOVI). Visual space is sampled according to a mathematical model of the retino-cortical mapping (Rovamo & Virsu, 1984), as shown in Figure 1A; our novel k-nearest-neighbor (kNN)-convolution and kernel mapping method enables perceptual processing over this foveated input format. This sampling interface draws on known characterizations of the primate visual system: cutting this manifold on the long-axis – corresponding to the vertical meridian dividing left and right visual fields – yields a strong first-order match to the retinotopic organization of human V1.

2. We present a *novel foveated convolutional neural network* that natively learns convolutional features over foveated in-

put (Figure 1B). We train these models end-to-end, while varying the degree of foveation, and demonstrate both biologically-plausible spatial receptive field properties, and an advantage for intermediate foveation levels in image classification compared to non-foveated control models.

3. Third, we show how to *outfit a state-of-the-art pre-trained vision transformer with a foveated sensing interface* (Figure 1C). Our method uses the kNN-convolution to implement a foveated patch embedding into an otherwise standard vision transformer, and we explore fine-tuning protocols leveraging low-rank adaptation (LoRA) (Hu et al., 2021) to functionally integrate the new sensor into two sizes of DINOv3 ViTs. These models achieves high performance at a reduced computational budget, approaching the performance of full-resolution baselines, and beating matched non-foveated variants with the same limited resource constraints, while unlocking possibilities for efficient active sampling in high resolution settings.

# 4. A biologically-inspired foveated vision interface

In standard computer vision models, the representation of the visual world is rectangular (the size of the image), and the model can perform regular (i.e. convolutional) processing directly over the input image using rectangular kernels or patches (e.g. 3x3 pixel) that evenly tile the activation map. This regular processing relies on the presence of a regular grid representation of the image.

However, we seek to sample points in a foveated manner where resolution depends only on eccentricity – but not polar angle; this set of points can be reformatted into a uniformly dense, but curved, manifold (Figure 1A). We thus introduce the foveated vision interface (FOVI), containing two components: 1) a foveated sensor with non-uniformly dense sampling coordinates and a uniformly dense sensor manifold, and 2) a mechanism for regular processing on the sensor manifold.

## 4.1. Foveated sampling with a uniformly-dense sensor manifold

The mathematical model of (Rovamo & Virsu, 1984) describes an exact uniform representation of an array of points whose sampling density depends only on eccentricity, which is a strong first-order approximation of human foveated sensing, and it's corresponding spatial organization in primary visual cortex (V1). The sampling density is determined by the "cortical magnification function" (CMF; Daniel & Whitteridge, 1961) – so called because it describes the magnification of visual space in V1 of monkeys and humans – for which we use the standard form $M(r) = \frac{1}{r+a}$ (2A). Integrating the CMF yields the cortical distance along the

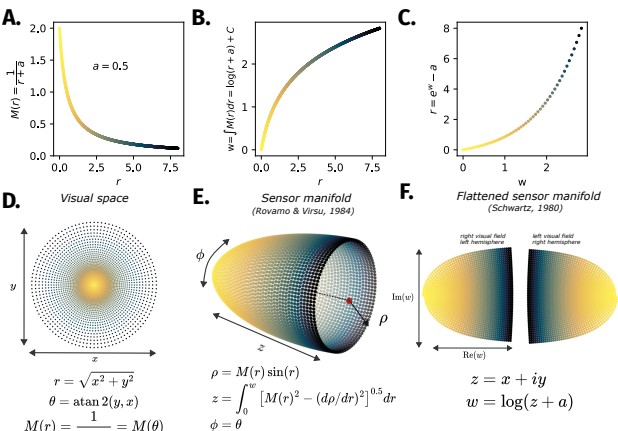

**Figure 2.** The relationship between cortical magnification and isotropic foveated sensing. **A.** The cortical magnification function commonly used to account for the organization of retinotopic maps in visual cortex (Van Essen et al., 1984; Schwartz, 1994). We set $a = 0.5$ and a field-of-view of 16 degrees. **B.** The integral of the CMF $w$, from $0$ to $r$, yielding the "cortical" dimension corresponding to eccentricity. **C.** Sampling evenly along the domain of $w$ and solving for the corresponding retinal radius $r$ to achieve foveated samples in visual space. **D.** Sensor locations in visual space arising from isotropic foveated sampling. **E.** Visual points from **D.** mapped in the manifold of Rovamo & Virsu (1984). **F.** Visual points from **D.** mapped into the complex log model (Schwartz, 1980).

cortical axis aligned with eccentricity (Figure 2B), here, $\log(r + a) + C$. Finally, uniformly sampling the range of cortical distance yields a set of radius values (Figure 2C) that are equally spaced on the sensor manifold. We then determine the number of equally spaced angular samples to select in visual space in order to preserve local isotropy; that is, to ensure that the distance between neighboring angles is equal to the distance between neighboring radii at any given point. This ensures locally consistent spacing (local isotropy) throughout the visual field, while achieving magnification along the radial dimension (Figure 2D). Together, this sampling strategy produces points that are evenly distributed on the 3D sensor manifold (Figure 2E). Due to our choice of magnification function, the "complex log" model of Schwartz (1980) can be used as an ideal "flat" representation that allows for 2D visualization of the entire sensor manifold–a cut is made along the vertical meridian, and the two hemifields are flattened, mirroring the spatial organization of visual information in the hemispheres of area V1 (Figure 2E).

To control the *strength of foveation* in our models, we vary the $a$ parameter in the CMF. With the CMF $M(r) = \frac{1}{r+a}$, as $a \to \infty$, $M(r)$ approaches a constant value, yielding uniform sampling. Thus, smaller values of $a$ indicate stronger foveation, or less uniform sampling across eccentricity. As shown in Figure 2C, a desired number of visual sampling radii ($n_r$) are generated from equally spaced samples of the

integrated CMF (or log radius, $\log(r+a)$), from the minimum to maximum value; given $n_r$, the number of samples $n_s$ is then determined via enforcing local isotropy, rather than being directly specified. For a given $a$, we thus search over the number of radii $n_r$ that produces the closest match to a target number of samples $n$, ensuring approximately matched resources across models.

## 4.2. Kernel mapping for kNN-convolution on the sensor manifold

To perform perceptual processing on the sensor manifold, we specify spatial receptive fields as k-nearest-neighborhoods (kNNs) around a set of output units tiled across the same manifold. To enable convolutional weight-sharing across kNNs on the manifold, we introduce the **kNN-convolution**, mediated by a novel **kernel mapping** technique, which maps a reference kernel ($W$) – learned in a standard Cartesian grid – into each neighborhood (Figure 3). A goal of this mapping is to achieve convolutional kernels that are aligned across locations in visual space, such that a vertical orientation filter detects vertical features across the entire image. To do so, we define a set of polar neighborhood coordinates for each kNN, setting the radius $r$ as the geodesic distance from the output unit on the sensor manifold, and the polar angle $\theta$ as the polar angle in *visual* space, computed with respect to the output unit (Figure 3, step 1). We then determine the Cartesian neighborhood coordinates using the standard formulas $x = r\cos(\theta)$, $y = r\sin(\theta)$, achieving aligned visual angles (Figure 3, step 2); these coordinates lie in the same frame as the learned reference kernels ($W$). Finally, we spatially sample the reference kernel with each neighborhood using the reference frame coordinates, shown in Figure 3, step 3. Plotting kernels back into the flattened sensor space highlights their uniform size on the sensor manifold; projecting the learned feature kernel back into visual space shows how the learned filter is the same across visual space, with different visual field coverage depending on eccentricity.

The reference kernel resolution can be set at the same resolution of the kNNs (square side length $s = \sqrt{k}$), or at a higher-resolution to better accommodate idiosyncratic spatial positions across kNNs (c.f. anti-aliasing). In practice, we find that using a higher-resolution reference kernel ($s = 2\sqrt{k}$) leads to stronger performance, improving ImageNet-1K accuracy by approximately 3% (see Figure S9). We make this our default. Notably, while this does increase parameter count, it is heavily constrained by the fixed spatial mapping, and each mapped kernel still uses the same number of parameters as a standard 2D kernel of side length $s = \sqrt{k}$.

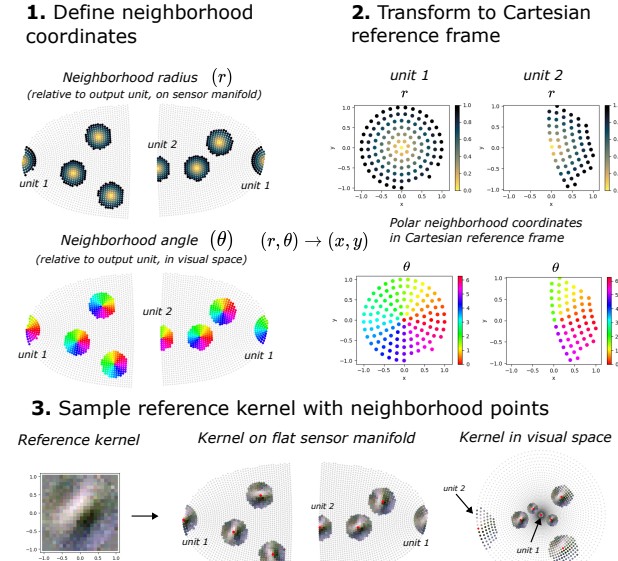

**1.** Define neighborhood coordinates

*Neighborhood radius* $(r)$
*(relative to output unit, on sensor manifold)*

**2.** Transform to Cartesian reference frame

*Neighborhood angle* $(\theta)$    $(r,\theta) \rightarrow (x,y)$
*(relative to output unit, in visual space)*

*Polar neighborhood coordinates in Cartesian reference frame*

**3.** Sample reference kernel with neighborhood points

*Reference kernel*    *Kernel on flat sensor manifold*    *Kernel in visual space*

**Figure 3.** Kernel mapping procedure for the kNN-convolution operation. First, even k-sized neighborhoods are defined in the sensor manifold (upper), along with their orientation in the visual input (cartesian) space (lower). Second, kernels are transformed into a common cartesian reference frame, and aligned to a common reference kernel. Third, we visualize the learned reference kernel across different units in both the flattened representational space (top), and in visual space (bottom). These show how the kernel is a fixed size in the sensor manifold, and orientionally-aligned in visual cartesian coordinates, while scaling with eccentricity.

## 5. FOVI-CNNs

Given this sensor interface, we next illustrate the development of a foveated convolutional architecture, which leverages multiple layers of kNN-convolution operations, supporting hierarchical foveated feature learning; we call these networks FOVI-CNNs. Each layer's activation map is formatted as a sensor manifold, with a particular resolution of evenly spaced samples. The resolution of each layer decreases progressively through the network, as in typical CNNs. Each layer defines the kNN centers for processing of the previous layer (Figure 1B).

To instantiate a particular FOVI-CNN architecture, we paralleled the choices of a simple AlexNet-like (Krizhevsky et al., 2012) convolutional model, with 5 convolutional layers, 3 pooling layers, and 2 fully connected layers. The first convolutional layer uses a kernel size of 11 and stride of 4, the second convolutional layer uses a kernel size of 5 and stride of 1, and the remaining convolutional layers use a kernel size of 3 and stride of 1. Besides the initial sampling layer, downsampling is performed strictly in (3x3) max pooling layers with stride 2, following the first and fourth convolutional layers. Before the first fully connected layer, a global average pooling layer is used, as in ResNets and other later architectures (He et al., 2016). Across con-

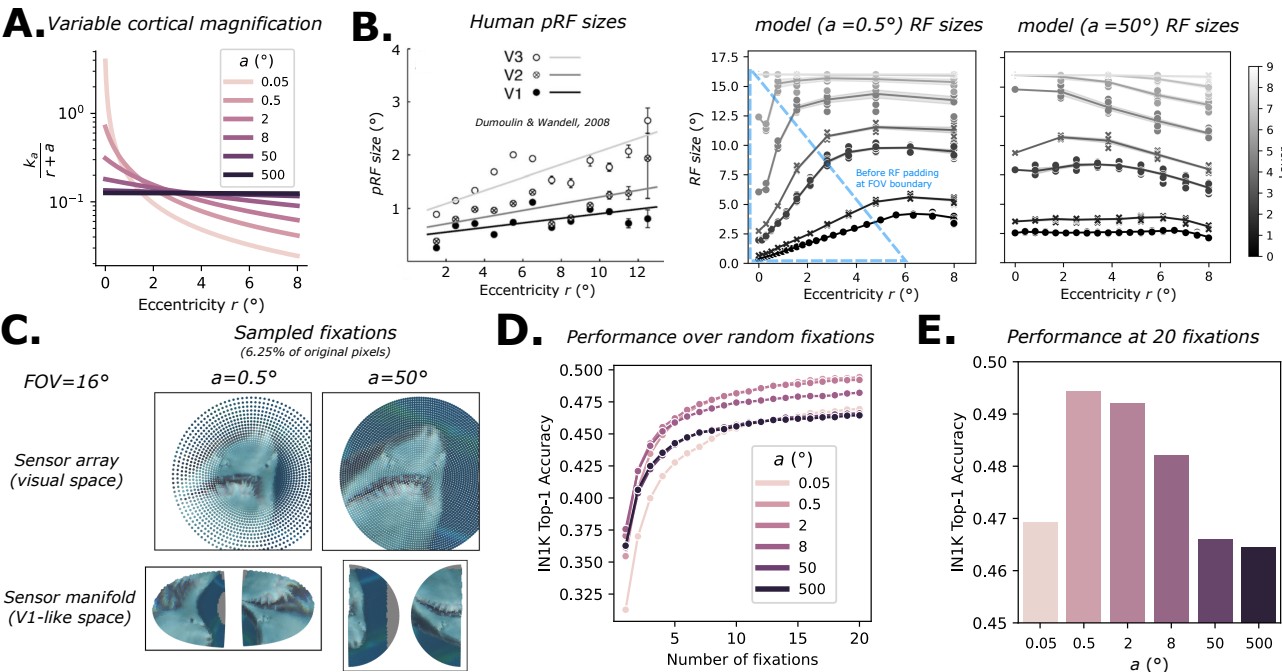

**Figure 4.** FOVI-CNNs account for the spatial characteristics of primate neural receptive fields, and provide a performance advantage in ImageNet classification. **A.** Implementing variable foveation via variability in the cortical magnification function (CMF). Given the cortical magnification function (CMF; $M(r)$), we can specify a continuum of foveation, where small $a$ corresponds to strong foveation, and uniform sampling is achieved as $a \to \infty$. Note: we scale the CMF by $k_a$, where $k_a = (\int_0^{r_{max}} \frac{1}{r+a}\,dr)^{-1}$, in order to normalize the area-under-the-curve across models. **B.** Left: human population receptive field (pRF) sizes, measured with fMRI (Dumoulin & Wandell, 2008). Middle: receptive field (RF) sizes across layers in a foveated model ($a = 0.5$). The blue box shows an approximate region in which RF sizes scale linearly with eccentricity, before padding begins to take place due to reaching the FOV boundary; padding units are not included in the RF size calculation. Right: RF sizes in a nearly uniformly sampling model ($a = 50$). **C.** Sampled fixations for strong and weak foveation models, shown both as a sensor array in visual space, and a (flat) sensor manifold in a V1-like space. **D.** ImageNet-1K results after training, mapping out performance up to 20 random fixations. **E.** Highlight of performance across foveation levels at the maximum of 20 fixations.

volutional layers, there are 96, 256, 384, 384, and 256 channels. We implement padding by extending the sampling outside the processing field-of-view, labeling such units as padding units whose activation always maps to 0. In the formation of kNNs in the following layer, these padding units are then automatically selected as nearest neighbors to appropriately pad the input; "unit 2" in Figure 3 is an example unit whose receptive field is padded. Finally, we add two fully-connected layers with 1024 units, use a ReLU nonlinearity, and perform batch normalization after each nonlinearity, with a learned affine transformation (Ioffe & Szegedy, 2015).

### 5.1. FOVI-CNNs match the spatial characteristics of primate neural receptive fields

First, we demonstrate that this model architecture produces a strong match to the spatial characteristics of primate neural receptive fields. In particular, receptive field mapping studies have consistently demonstrated that primate receptive fields are larger the more peripheral they are, and the farther

up the hierarchy they are (Dumoulin & Wandell, 2008; Motter, 2009). Figure 4A depicts this phenomenon in human V1-V3, showing a linear increase in "population receptive field size" (pRF) size with eccentricity, with an increase in both slope and intercept across hierarchical areas.

We characterized the spatial RF properties in each layer of our model, assessing both a foveated ($a = 0.5$) and non-foveated ($a = 50$) variant. For each layer, we compute the RF diameter of every feature map location by back-projecting receptive field neighborhoods through each preceding layer until reaching the input layer. We then plot the RF diameter as a function of eccentricity for each layer. As shown in Figure 4B, middle, we see an approximately linear increase in RF size with eccentricity. This linear increase eventually plateaus for each layer once the receptive field centers get far enough into the periphery to incorporate padding units beyond the field-of-view, such that the RF does not continue to grow. Additionally, both the slope and intercept of RF diameter by eccentricity functions increase with the hierarchical layer number. The dependence of RF size on eccentricity, but not hierarchical layer, falls directly

out of the foveated sampling, as it is abolished when assessing the non-foveated variant, shown in Figure 4B, right. Beyond the qualitative patterns of RF sizes across visual areas, foveated sampling allows us to account for the precise shapes of primate receptive fields, unlike log-polar and warped cartesian approaches, which both warp receptive fields due to anisotropic sampling (see Appendix 10.2).

## 5.2. Moderate foveation boosts classification accuracy

Next, we examine the consequences of foveation for perceptual performance in image classification. Recall that the motivation behind foveation is that it is useful when operating over a much higher resolution input than allowed by the sensor, as in the ambient light field in the real world. Here, we simulated this scenario by giving FOVI-CNNs a constrained "pixel-budget" (64x64) to sample from a 256x256 "ambient" resolution, representing a 16-fold reduction in samples. Examples of foveated samples can be seen in Figure 4C. (Note that since these models were constrained to only sample $64^2$ pixels, we replaced the stride of 4 in the first layer with a stride of 2, in order to prevent feature map resolution from shrinking to 1 before the final convolutional layer.)

We perform experiments using the ImageNet dataset (Russakovsky et al., 2015). For more detailed experimentation across hyperparameter variations, we also use a smaller custom ImageNet subset which we refer to as ImageNet-100. This dataset consists of 100 random ImageNet categories, and contains the same number of images as CIFAR-100 (500 training images per category, and 100 validation images per category); basic validations in result trends across datasets are shown in Figure S5. To facilitate faster training, both datasets are re-sampled to a uniform maximum resolution of 256 for use with FFCV (Leclerc et al., 2023), unless otherwise mentioned. We train for 100 epochs, using a cosine-decay learning rate scheduler, without early stopping.

We train our models using 4 random fixations drawn from a central area of the image; in our main experiments we use a radius of 0.25 of the image size to define the fixation zone, however we also experiment with a larger radius of 0.45 (Figure S8). Due to gradient accumulation, it is expensive to train on many fixations, imposing a similar cost (and benefit in learning) to increasing the number of epochs. However, as inference is computationally cheaper and thus more amenable to large numbers of fixations, we allow for up to 20 random fixations during validation. We forego crops, as cropping is a form of foveation that focuses perception on a restricted part of the image at higher resolution than would otherwise be allowed; we examine this choice in Figure S7. Each fixation produces independent logits for classification. We adopt a simple aggregation strategy, averaging

logits across fixations before computing the final prediction. During training, the average logits are used with the cross-entropy loss for supervised learning; during validation, these average logits are used to generate top-1 and top-5 accuracies. In pilot analyses, we found this simple aggregation performed as well as or better than more learned recurrent integrators, and thus adopted it for simplicity. Plotting performance over the number of fixations (Figure 4D), we see that each model improves significantly with increasing fixations, generally saturating in improvement by about 20 fixations. Focusing in on the performance at the maximum 20 fixations (Figure 4E), we see an inverted U-shaped function over the foveation parameter $a$, with peak performance at an intermediate foveation level of $a = 0.5$; critically, models with intermediate foveation showed better image recognition than the pixel matched uniform model ($a = 500$). We discuss these results in further detail, along with a series of follow-up analyses, in Appendix 10.4; briefly, it seems this degree of foveation is well suited to capture the center-bias and scale-bias of the critical object content in these ImageNet images. These results demonstrate the successful application of our foveated sensor and kNN-convolution, highlighting an example performance benefit for foveation under pixel resource constraints.

## 6. FOVI-ViTs

We next show how the foveated interface can be integrated into transformer architectures, including models that have been pre-trained on extremely largescale data.

### 6.1. Foveated patchification through kNN-convolution

The key to connecting the foveated interface to a transformer architecture is developing a suitable patchification scheme. We leverage our kNN-convolution. Precisely, we define two foveated sampling grids: 1) a sensor array, and 2) a patch-center array. Patches are defined as kNNs over the sensor array, using distances on the sensor manifold as typical in the kNN-convolution. We choose the length of the patch-center array to exactly match the number of patches in a baseline ViT, by constraining the set of $a$ values to those that produce the desired number of patches; this procedure is explained further in Appendix (10.7). For $n = 64$ patches, we determine 5 suitable $a$ values (rounded here to two decimal places): 0.03, 0.14, 0.58, 2.79, 60.94. We set the patch size $k$ to the minimum such that all sensor locations are included in at least one patch-center kNN; due to the circular nature of kNNs, this induces a small degree of overlap, similar to the use of overlapping convolutions in some vision transformer variants (e.g. Xiao et al., 2021). Patchification is shown in Figure 1C.

Broadly, this interface allows ViT architectures to process the image in a foveated manner, with more tokens dedicated

| | # fix. | pixels | patches | GFLOPS | acc. [val] | lat. (ms) [train] | lat. (ms) [val] | mem. (GB) [train] | mem. (GB) [val] |
|---|---|---|---|---|---|---|---|---|---|
| ViT-H+ uniform @ 224 | 1 | 50176 | 196 | 172.39 | 0.871 | 289.59 | 103.80 | 40.1 | 4.1 |
| FOVI-ViT-H+ @ 64 (a=2.79) | 1 | 3976 | 64 | 58.43 | 0.844 | 119.95 | 45.01 | 19.1 | 4.1 |
| FOVI-ViT-H+ @ 64 (a=2.79) | 3 | 11928 | 192 | 175.30 | 0.853 | 303.70 | 111.76 | 40.9 | 4.2 |
| ViT-S+ uniform @ 224 | 1 | 50176 | 196 | 6.16 | 0.794 | 37.92 | 15.53 | 4.1 | 1.0 |
| FOVI-ViT-S+ @ 64 (a=2.79) | 1 | 3976 | 64 | 2.04 | 0.700 | 27.61 | 10.84 | 1.7 | 1.0 |
| ViT-S+ uniform @ 64 | 1 | 4096 | 64 | 2.02 | 0.693 | 27.60 | 10.79 | 1.7 | 1.0 |
| Weak FOVI-ViT-S+ @ 64 (a=60.94) | 1 | 4032 | 64 | 2.04 | 0.693 | 27.41 | 10.76 | 1.7 | 1.1 |
| ViT-S+ log-polar @ 64 (a=2.79) | 1 | 4096 | 64 | 2.02 | 0.643 | 27.86 | 10.77 | 1.7 | 1.0 |
| FOVI-ViT-S+ @ 64 (a=2.79) | 3 | 11928 | 192 | 6.12 | 0.735 | 46.78 | 23.63 | 4.3 | 1.1 |
| ViT-S+ uniform @ 64 | 3 | 12288 | 192 | 6.06 | 0.726 | 46.66 | 23.47 | 4.2 | 1.1 |
| Weak FOVI-ViT-S+ @ 64 (a=60.94) | 3 | 12096 | 192 | 6.12 | 0.725 | 46.76 | 23.59 | 4.3 | 1.1 |
| ViT-S+ log-polar @ 64 (a=2.79) | 3 | 12288 | 192 | 6.06 | 0.694 | 46.89 | 23.54 | 4.2 | 1.1 |

**Table 1.** Performance comparison of DINOv3 variants. The table is divided into H+ and S+ variants, then sorted by number of fixations, and then sorted by accuracy. GFLOPs are reported per image, while latency and memory are computed in both training and validation modes with a batch size of 64, using a single NVIDIA H100 GPU. In training mode, both forward and backwards passes are run; in validation mode only a forward pass is run, with gradients disabled. "@ 64" indicates using a sampling resolution parameter of 64, equivalent to approximately $64^2$ pixels, and $a$ refers to the foveation parameter (smaller = stronger foveation). Uniform full-resolution baselines are run only for 1 fixation. Abbreviations: accuracy (acc.), fixation (fix.), latency (lat.), peak memory (mem.).

to the center of gaze and progressively less dedicated to the periphery, dependent on the magnification parameter $a$. Compared to CNNs, the ViT architecture only requires a single kNN-convolution for patch embedding, rather than one per layer. If the ViT is pre-trained, given the kNNs defined by the patchification, it is also possible to make use of pre-trained patch embeddings as the reference kernel in kNN-convolution, which is then mapped into the patches using the standard kernel mapping procedure (Figure 3). Note, while in this work we focus on adapting pre-trained ViTs, FOVI can also be used to train ViTs from scratch or via distillation with the same simple replacement of the patch embedding layer. The lack of inductive bias in ViTs leads to poor from-scratch supervised performance on ImageNet, requiring distillation (Touvron et al., 2020), strong data augmentation and regularization (Steiner et al., 2021), or pretraining to achieve strong performance. Here we focus on the role of pretraining, adapting foveated ViTs from the large-scale pretrained DINOv3 model.

### 6.2. Leveraging pre-training for foveation via adaptation

We next outfit the pre-trained DINOv3 model (Siméoni et al., 2025) with a foveated interface. We focus on the small but performant ViT-S+ variant, and additionally assess the larger and more expensive ViT-H+ variant. In pilot experiments, we determined a suitable strategy for adapting DINOv3 to work with foveated inputs. This strategy uses LoRA (Hu et al., 2021) over the patch embedding and first half of the network, which significantly reduces overfitting relative to full fine-tuning, better retaining the visual feature representations acquired during pre-training while adapting to the foveated inputs. We describe our process for determining

this strategy in Appendix 10.6; briefly, on IN-100, LoRA over the patch embedding and first half of transformer layers outperforms frozen adaptation by $\sim 30\%$, full fine-tuning by $\sim 10\%$, and late-only adaptation by $\sim 15\%$.

We train FOVI-ViT-H+ and FOVI-ViT-S+ models with a resolution parameter of 64 and patch size parameter of 8, targeting 64 patches. For exact patch count specification, only a discrete set of $a$ values is allowed (see Appendix 10.6 for tuning analyses). We select $a = 2.79$, which achieves moderate foveation. ViT-S+ models were trained for 100 epochs and ViT-H+ models were trained for 25 epochs, both using a cosine decay learning rate scheduler without early stopping; ViT-H+ was trained for a shorter duration due to its faster convergence (in number of epochs) and more expensive per-epoch training time.

We compare FOVI-DINOv3 models to four baselines. The first is the original, full-resolution model, run with 1 fixation (**uniform @ 224**). The second is a matched **weak-FOVI @ 64** baseline, using $a = 60.94$ to minimize foveation, while retaining the same kNN-convolution-based patch embedding method. The third is a **uniform @ 64** baseline, where the image has been downsampled to 64x64 to match the number of pixels sampled by the foveated variant, and kernels are similarly downsampled from 16x16 to 8x8, allowing for a direct comparison with standard patch embedding but approximately matched resources. The last baseline is a matched **log-polar @ 64** baseline, in which images are sampled according to the Log-Polar foveation model (e.g. Javier Traver & Bernardino, 2010) at the desired resolution, and then are otherwise processed identically to the uniform baseline. Additionally, we explore ViT-S+ models trained with a different number of pixels, but the same total number of patches (via larger patch sizes) in the Appendix

(10.8). We use a matched LoRA strategy for low-resolution baselines, and freeze the high resolution baslines. Low-resolution models are trained using 4 random near-center fixations, and evaluated using 1 or 3 fixations, using an identical protocol to that used for CNNs in the previous section, but constraining to a smaller number of evaluation fixations to keep GFLOPS approximately less than or equal to the high-resolution uniform baseline. Results are reported in Table 1.

Notably, using only 1/16 of the pixels and approximately 1/3 of the total GFLOPS, our FOVI-ViT-H+@64 variant achieves > 96% of the accuracy of the ViT-H+ Uniform@224 baseline in a single fixation, setting what is to our knowledge the state-of-the-art performance on ImageNet-1K at a resolution of $\leq$ 4096 pixels (84%). The smaller FOVI-ViT-S+@64 variant achieves 88.2% of the accuracy of the ViT-S+ Uniform@224 baseline in a single fixation. In addition, our FOVI models beat the other low-resolution baselines, including Weak-Fovi @ 64 ($a = 60.94$), Uniform @ 64, and Log-polar @ 64 baselines. The performance is notably better than log-polar, which likely struggles to be adapted due to lack of consistent filter orientation in log-polar images, and thus a stronger distribution shift from pre-training. In terms of efficiency, we find that 1-fixation FOVI achieves significantly reduced GFLOPS, latency in both training and validation modes, and memory during training mode; in validation mode, memory costs of the forward pass are largely outweighed by the model size, since intermediate activations can be discarded along the feedforward path when gradients are disabled. Despite the increases in efficiency, typically the reductions in latency and memory are somewhat less pronounced than the reduction in FLOPS; this is dependent on various aspects of hardware, model size, and batch size, which cannot be fully explored here. In sum, these results demonstrate that FOVI can be used to adapt vision foundation models, and achieve competitive performance at reduced computational cost, particularly in the larger ViT-H+ variant.

### 6.3. Efficiency at high resolution

Last, we briefly analyze the efficiency implications of FOVI for high-resolution scenarios. First, we analyze the GFLOPs/image in DINOv3 (ViT-S+), breaking down the GFLOPs into attention-based processing, and non-attention-based processing (Figure 5A). We define the image resolution in terms of the side length of a Cartesian image ($m = \sqrt{n}$, where $n$ is the number of samples taken in a resource-matched foveated model. Since image resolution scales as $O(m^2)$, we find empirically that the non-attention (mostly linear) operations are approximately quadratic ($O(m^{1.76})$), whereas the attention-based operations scale exactly with $O(m^4)$ due to their quadratic dependence on sequence length ($O(n^2)$). For $m < 400$, the cost of non-attention

operations (i.e. linear layers, nonlinearities, normalizations, etc.) outweighs the cost of attention-based operations. However, for larger resolutions, the attention-based cost becomes enormous. This illustrates the inherent difficulty of scaling transformers to high-resolution inputs.

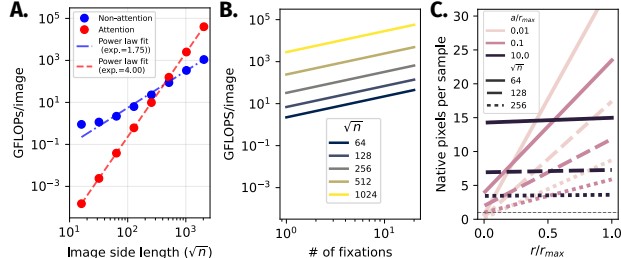

**Figure 5.** Analyzing efficiency in ViT-S+ from the lens of foveation. **A.** GFLOPs/image as a function of image resolution, separately for attention and non-attention operations. Power laws are fit to each curve as empirical $O(m)$ analyses, where $m = \sqrt{n}$ is the pixels per side of a square image. **B.** GFLOPs/image as a function of the number of fixations, for different image resolutions. **C.** Local sampling resolution (native pixels per sensor sample) as a function of eccentricity, varying the resolution and foveation of the sensor. A horizontal line at 1 indicates the native resolution.

Figure 5B shows GFLOPs per image as a function of fixation count for different sensor resolutions ($m = \sqrt{n}$). Even with 20 fixations, processing at $\sqrt{n} = 64$ requires two orders of magnitude fewer GFLOPs than a single pass at $\sqrt{n} = 1024$. Figure 5C illustrates the underlying trade-off: stronger foveation (smaller a) achieves native sampling resolution at the center of gaze, but at the cost of coarser peripheral sampling. Practitioners can tune this trade-off based on task demands, or perhaps more promisingly, use reinforcement learning to optimize the visual behavior of active agents (viewing distance or crop size and fixation patterns) depending on their tasks and sensor characteristics.

## 7. Conclusion

For decades, researchers have sought a mechanism to mimic the human visual system's remarkable efficiency via computational approaches for foveation (e.g. Weiman & Chaikin, 1979; Schwartz, 1980; Basu & Licardie, 1995), with a recent resurgence in interest (Wang et al., 2021; Jérémie et al., 2024; Deza & Konkle, 2020; Kerr et al., 2025; Chuang et al., 2025; Da Costa et al., 2024; Killick et al., 2023). Our work is the first to accurately model the locally isotropic and eccentricity-dependent nature of human foveation, in a general approach that can be applied across any vision architecture. FOVI thus unlocks a flexible pipeline for exploring resource constraints in deep vision models by controlling the degree of foveation, with uniform sampling being a special case ($a \rightarrow \infty$). Our experiments with FOVI-ViTs demonstrate practical utility, with our FOVI-ViT-H+ variant

achieving state-of-the-art accuracy in the low-pixel regime (84% ImageNet top-1). Thus, the primary contribution of this work is in the design of the foveated sensor interface and its integration into vision architectures via kNN convolution and low-rank adaptation.

## 8. Limitations

One limitation is that the kNN convolution introduces some computational overhead in processing relative to standard 2D convolution. This is because - while all kNNs derive their weights from the same convolutional reference frame - each kNN has a different scattered layout, so it has proven more challenging to optimize vs. the more regular 2D convolution. This is more problematic for CNNs – which pay this cost at every stage – compared to ViTs, which pay the cost only once at the patch embedding stage. Future work should seek to optimize kernels to minimize the computational overhead of kNN convolution. Another limitation is that we found kNN convolution to work best when at least some of the layers have reference frames with $s^2 > k$. This increases parameter count, and (depending on the particular implementation)can increase memory usage in the forward pass. While this concern is most relevant for CNNs, it warrants further exploration for particular use cases. Our open-source toolbox exposes multiple kNN convolution backends, and more optimizations will be attempted. Finally, our evaluations are limited to image recognition in a low-resolution setting where results are modest, and do not yet demonstrate the ability of FOVI to serve object detection or segmentation; moreover, we do not rigorously compare FOVI with non-foveated generic patch reduction techniques (e.g. Liu et al., 2021; 2022). We consider foveated perception distinct from pure patch reduction, as it also limits the number of pixels sampled, which has additional benefits.

## 9. Future directions

Foveation may be more beneficial in settings where higher resolution processing is demanded and thus computational constraints are of greater concern, such as large field-of-view naturalistic scenes (Shi et al., 2025), or interaction within artificial or natural high-resolution worlds (Yu et al., 2025; NVIDIA et al., 2025). As we showed in Figure 5, while low resolutions exhibit locally-linear scaling, higher resolutions exhibit more locally-quadratic scaling, as the attention operations begins to outweigh the linear layers; this creates a stronger need for patch reduction techniques such as foveation (c.f. Shi et al., 2025; 2026). Similarly, video processing poses stronger needs for patch reduction, and can benefit from spatiotemporal redundancy (Shi et al., 2026). Second, to make foveation practical for real-world tasks, advances are needed in mechanisms both for active vision and saccadic integration, as implemented in prior

works exploring simplified mechanisms for foveation (e.g Elsayed et al., 2019; Jonnalagadda et al., 2022). The inherently temporal, interactive, and embodied nature of robotic vision strongly motivates further exploration of foveation in robotics, as do notable recent successes learning active vision policies in robotics with simplified foveated architectures (Kerr et al., 2025; Chuang et al., 2025) and even non-foveated models that must explore partially observed environments (Luo et al., 2025). Third, while here we explore the *perceptual* efficiencies of foveation, restricting the number of samples can also increase *sensing* efficiency in multiple applications. Within robotics simulation environments such as IsaacLab (NVIDIA et al., 2025), the foveated sensor can constrain ray-tracing during rendering, reducing the compute required to render each frame. Future camera hardware could also reduce energy and bandwidth costs by directly sampling in a foveated array to achieve high peak resolution with a small number of sensor elements. Finally, beyond computer vision applications, FOVI models holds promise for computational modeling of human vision, including modeling features of peripheral vision such as crowding (Pelli, 2008; Balas et al., 2009), how peripheral vision guides saccades for foveal viewing (Findlay & Walker, 1999; Zhaoping, 2024) and how peripheral vision supports other real-world behaviors (Vater et al., 2022). Technical applications (e.g. AR/VR) may use FOVI to efficiently render stimuli for human perception (c.f. Patney et al., 2016). To benefit the many possible future directions, we share an extensible `fovi` toolbox[†] and pretrained models[‡].

## Impact statement

This paper presents work whose goal is to advance efficient visual processing in machine learning, via inspiration from the human visual system. There are many potential societal consequences of our work, none of which we feel must be specifically highlighted here.

## Acknowledgments

We acknowledge funding from NSF CRCNS Award 2309041. We thank Gavriel State, Ruth Rosenholtz, Binxu Wang, and the Harvard Vision Sciences lab for productive feedback and discussions.

---

[†] https://github.com/nblauch/fovi
[‡] https://huggingface.co/fovi-pytorch

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

# 10. Appendix

## 10.1. Trade-offs in resolution, field-of-view, and processing resources

Figure S1 illustrates in detail the calculations we made regarding trade-offs in peak resolution, field-of-view, and processing resources.

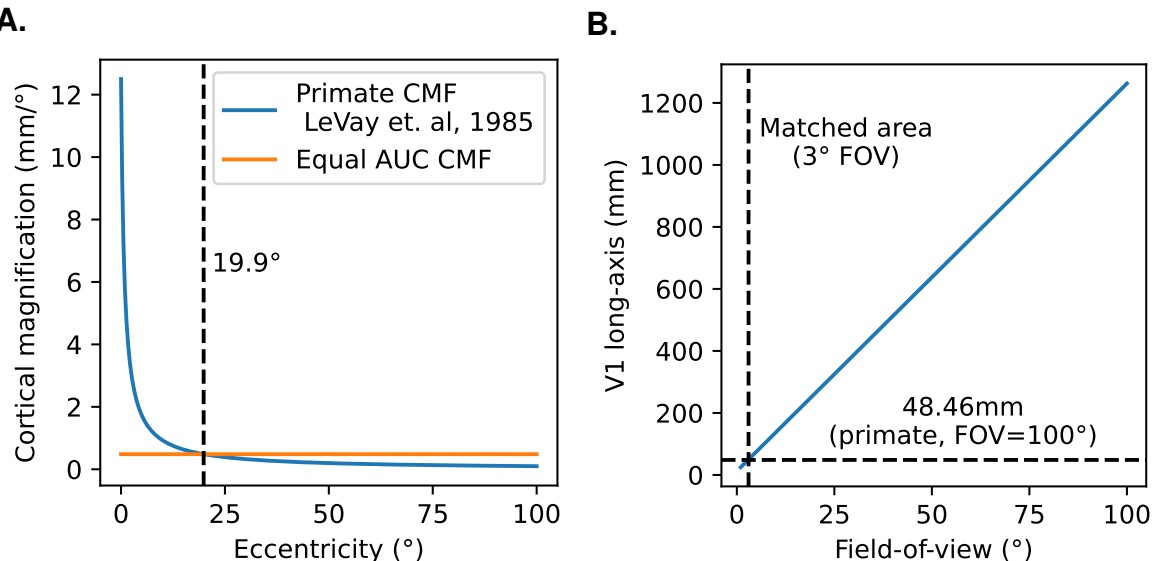

**Figure S1.** Illustration of trade-offs inherent to foveated vision. **A.** Cortical magnification function (CMF) from LeVay et al. (1985), used in our calculations, along with an equal area-under-the-curve uniform CMF. The uniform magnification is comparable to 19.9° in the standard CMF, thus affording intermediate-peripheral vision across the visual field for equal V1 size. **B.** Assuming uniform cortical magnification equivalent to the central value (12.5 mm/deg), we solve for the length of V1 by integration, and indicate lines corresponding to matched V1 area and the allowable field-of-view under this uniform magnification.

## 10.2. Accounting for the shapes of primate neural receptive fields

In the main text, we demonstrate that our foveated FOVI-CNN accounts for the progressive increase in RF size with eccentricity in primate neural receptive fields. Here, we demonstrate that it also accounts for their shapes. We plot the data of Motter (2009) in Figure S2D. On the left, we see a contour plot of a V4 neuron's visual responses, showing the characteristic non-Gaussian shape, with an elongation of contours along the radial dimension. In the middle, this neuron's response profile is re-plotted as contours on the V1-like surface, demonstrating that the visual response pattern arises from approximately circular (or isotropic Gaussian) sampling on the V1-like surface. On the right, the isotropy is plotted by characterizing the ratio of short vs. long axis ratios in a bivariate anisotropic Gaussian fit over all measured neurons. The density of this distribution is heavily concentrated on a ratio of 1, demonstrating approximate isotropy with no systematic deviations from it. We refer the reader to Motter (2009) for further detail. In Figure S2E., we replicate these analyses in our foveated ($a = 0.5$) model. Given that our model is based on hierarchical isotropic sampling on the V1-like sensor manifold, it is no surprise that the receptive field properties follow the same trend as the macaque V4 neurons.

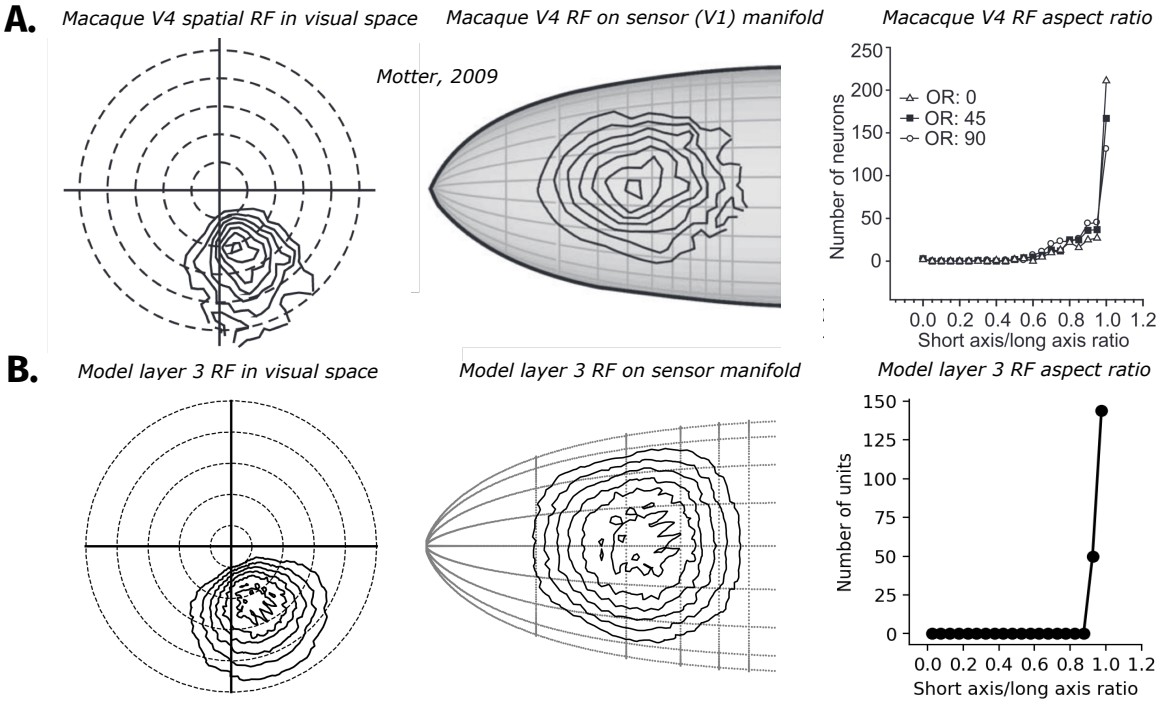

**Figure S2.** FOVI-CNN produces biological receptive field properties. **A.** Macaque V4 spatial RF from Motter (2009), plotted in visual (left) and sensor manifold space (middle). On the right, a histogram of aspect ratio is plotted across three model fitting orientations (see Motter (2009) for further detail). **B.** Example aggregated spatial receptive field from a unit in layer 3, plotting in visual space (left) and sensor manifold space (middle). On the right, a histogram of aspect ratio is plotted

## 10.3. Prior foveated sensors

**Cortical magnification and the complex logarithmic mapping function**     In this section, we discuss in detail prior work developing foveated sensors, and the issues inherent to them which we aim to address in the present work. Such work dates back at least to the pioneering work of Schwartz (1977; 1980), who demonstrated that a logarithmic mapping function could produce accurate fits of cortical magnification, mapping representation across the visual field to representation across primary visual cortex (V1). In key work, Daniel & Whitteridge (1961) defined the cortical magnification factor (CMF) as the amount of cortical space spanned by a fixed amount of retinal (or visual) space, finding that the CMF decreases sharply with eccentricity, but is roughly constant at all points in the visual field of constant eccentricity. This is known as *isotropic* cortical magnification, as the sampling rate is the same at a given point regardless of which direction it is measured. The search for a locally isotropic mapping function well matched to the empirical CMF, which is well fit by a function $M(r) = \frac{k}{r+a}$ (Van Essen et al., 1984, see Figure 2A.), led Schwartz to develop the complex logarithmic mapping model of V1 cortical magnification: $w = \log(z + a)$, where $z = x + iy$ is the complex plane (Figure 2E), and $\left|\frac{dw}{dz}\right|$ is the CMF. Along the horizontal meridian, this has the desired CMF $M(r) = \frac{1}{r+a}$, however, elsewhere the CMF is different due to the non-zero imaginary axis: $M(r) = \left|\frac{1}{x+a+iy}\right|$. Thus, while the CMF of the complex log model is locally isotropic, due to $a$ being a real constant, it exhibits a meridional anisotropy, in which the CMF along the vertical meridian (imaginary axis) is maximally different from that along the horizontal meridian (real axis), reaching a theoretical maximum ratio of $\sqrt{2}$ at $r = a$, at odds with empirical data (Himmelberg et al., 2023).

**The log-polar image model** As seen in Figure 2, the complex log model leads to a sensor manifold that is curved and disjointed at the vertical meridian, making computer vision applications difficult. A simplified version of this logarithmic mapping approach was thus developed, known as the log-polar mapping. This approach produces a grid-like image output, by simplifying the complex log: $\log(z + a) = \log(re^{i\theta} + a)$ as two dimensions of $\log(r + a)$ and $\theta$, where independent sampling can be done along each dimension. However, since an equal number of angles is selected at each radius, the resolution along the angular dimension is highly eccentricity-dependent, and this depends on the value of $a$. If $a = 0$, this approach would be correct, since $\log(re^{i\theta}) = \log(r) + i\theta$, that is, the complex plane of $r$ and $\theta$, matching the approach first developed by (Weiman & Chaikin, 1979). However, the log has a singularity at $a = 0$, and thus cannot be used in practice to model foveal eccentricities. Setting $a > 0$ removes the foveal singularity, but grid-sampling then introduces anisotropy. As a result, circular receptive fields drawn on a log-polar image with $a > 0$ can have very different shapes across various eccentricities. As $r$ increases, the aspect ratio of receptive fields increases in the tangential direction; for reasonably large values of $a$ (which can be helpful in not oversampling low to medium resolution images), this results in highly elongated peripheral receptive fields, in stark disagreement with empirical data (Motter, 2009). Thus, there is an inherent trade-off: at small values of $a$, the foveal magnification becomes extreme, but anisotropy is less severe, whereas at larger values of $a$, the foveal magnification becomes more realistic, but anisotropy becomes more severe. This approach has been used in several works, including recent work of (Gahl et al., 2024; Jérémie et al., 2024); for an earlier review see (Javier Traver & Bernardino, 2010).

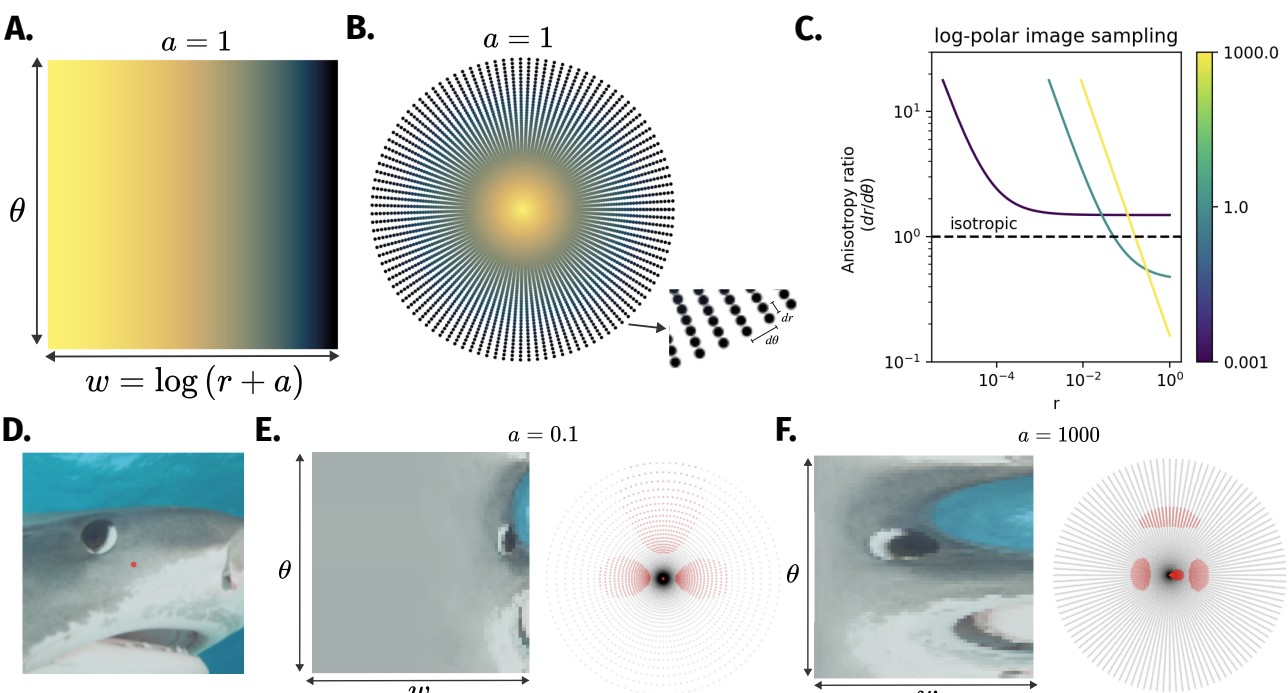

**Figure S3.** The log-polar image approach is anisotropic and introduces oversampling and warped receptive fields. **A.** The log-polar sensor manifold is a regular rectangular (here, square) grid of polar angle and log radius. **B.** The visual sampling induced by equally spaced points in the sensor manifold. Inset: illustration of anisotropy. $dr$ indicates the distance between neighboring radii, while $d\theta$ indicates the distance (arc length) between neighboring angles. **C.** The ratio $dr/d\theta$ is computed locally at each value of $r$ and plotted, for $a \in [0.0005, 0.5, 500]$. The dashed line corresponds to isotropic sampling. **D.** Illustration of log-polar foveation. Left: a target image with a central fixation point (red dot). Center: log-polar transformed image with $a = 0.05$. Right: log-polar transformed image with $a = 500$. **E.** Visual receptive fields corresponding to circular samples on the sensor manifold.

**Warped Cartesian approaches** Basu & Licardie (1995) introduced a related foveated sensing approach, also based on the complex logarithm (Schwartz, 1980), that yields a warped Cartesian image. However, as they note, this approach also leads to anisotropies in the periphery. A similar approach was used in recent deep learning approaches by Wang et al. (2021) and Da Costa et al. (2024), which both define a magnification function that depends only on eccentricity, though differing slightly from that inherent to the complex log. However, all of these models can be grouped into the family of warped cartesian image sensors, which show radially elongated receptive fields in the periphery. We illustrate the anisotropy and warping of receptive fields in Figure S4. Additionally, these models do not have a corresponding cortical space that can be mapped to visual cortex, foregoing some of the possible benefits in spatially relating activations in the model directly to brain responses.

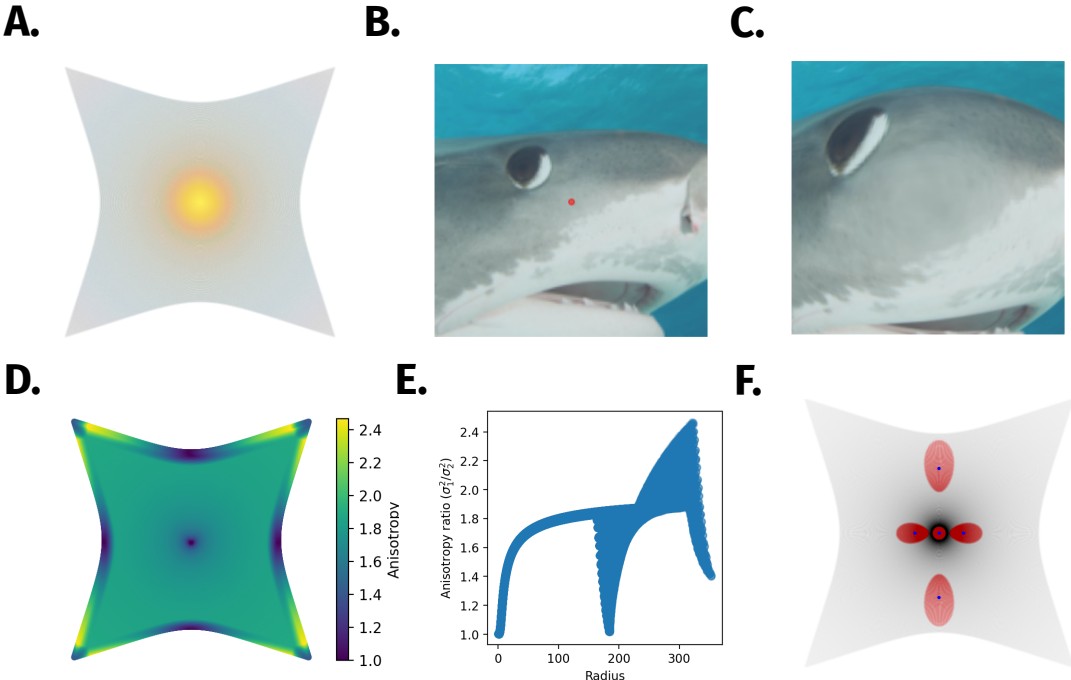

**Figure S4.** Warped Cartesian foveated sensors also introduce anisotropic sampling. Here, we use the sensor described in (Wang et al., 2021). **A.** Sensor locations. **B.** Target image with central fixation point (red circle). **C.** Warped Cartesian foveated image. **D.** Local anisotropy sampling plotted across the visual field. For this approach, at each point in the visual field, we sample $k = 1000$ (of $250^2$) nearest pixel neighbors. Then, we subject the pixel location matrix $X$ to an SVD to find the variance explained by the two principal directions of variance. Given $X^T X = U \Sigma V$, the variance explained by the first and second component are the squared entries of the diagonal matrix $\Sigma$, i.e. $\sigma_1^2$ and $\sigma_2^2$. We then compute the local anisotropy index as $\sigma_1^2 / \sigma_2^2$. **E.** Local anisotropy as computed in **D.**, plotted as a scatter plot against the visual field radius. **F.** Receptive fields drawn as circles in the foveated image space, projected back into visual coordinates.

**Log-Fibonacci sensor** Killick et al. (2023) introduce a foveated sensor that is most similar to ours, in that no attempt is made to wrangle the sensor outputs into an image, producing instead a point-cloud output that is designed to be input to a non-euclidean neural network. However, their model deviates from explicitly modeling cortical magnification in favor of simplicity. Their approach takes advantage of the golden ratio, to achieve sample packing that is approximately uniform within circles in retinal space, as in the seeds of a sunflower. However, while this sensor approach may be a reasonable choice for foveated computer vision, it does not have an associated cortical space, and thus has limited utility in modeling cortical organization, and is more difficult to tune, since it has multiple relevant hyperparameters. Additionally, the authors use structured Gaussian derivative-based filters for processing receptive fields encoded in this sensor space. This is expected to work well in the low to medium data regimes, but in the high data (non-foveated) regime where they were introduced, they were shown to perform worse than standard filters (Jacobsen et al., 2016). This low-data regime is that which was tested by Killick et al. (2023). Thus, it is unclear how well their approach would scale with increasing data. It would be possible to combine their sensor with our kernel mapping approach, however this is beyond the scope of our paper.

## 10.4. Exploring performance in FOVI-CNNs

To better understand our main FOVI-CNN results, we performed a series of follow-up analyses. Since it is computationally expensive to run many analyses, we first validate the use of a smaller subset of ImageNet, ImageNet-100, shown in Figure S5B. We find a similar pattern of validation accuracy across foveation levels, with peak performance for the intermediate foveation level of $a = 0.5$. Here, we additionally assess generalization by comparing accuracy on validation images with that on a matched subset of training images tested after training without data augmentation ("train-match-val"). First, examining ImageNet-1K, we find that more uniformly sampling models tend to overfit the data more, reflected by the increase in train-match-val accuracy without a corresponding increase in validation accuracy; this is seen particularly strongly for a smaller number of fixations. We see the same effect in ImageNet-100, albeit with a greater degree of overfitting. This suggests that foveation allows our models to avoid overfitting on less relevant background information, supporting stronger generalization.

**A.** IN-1K

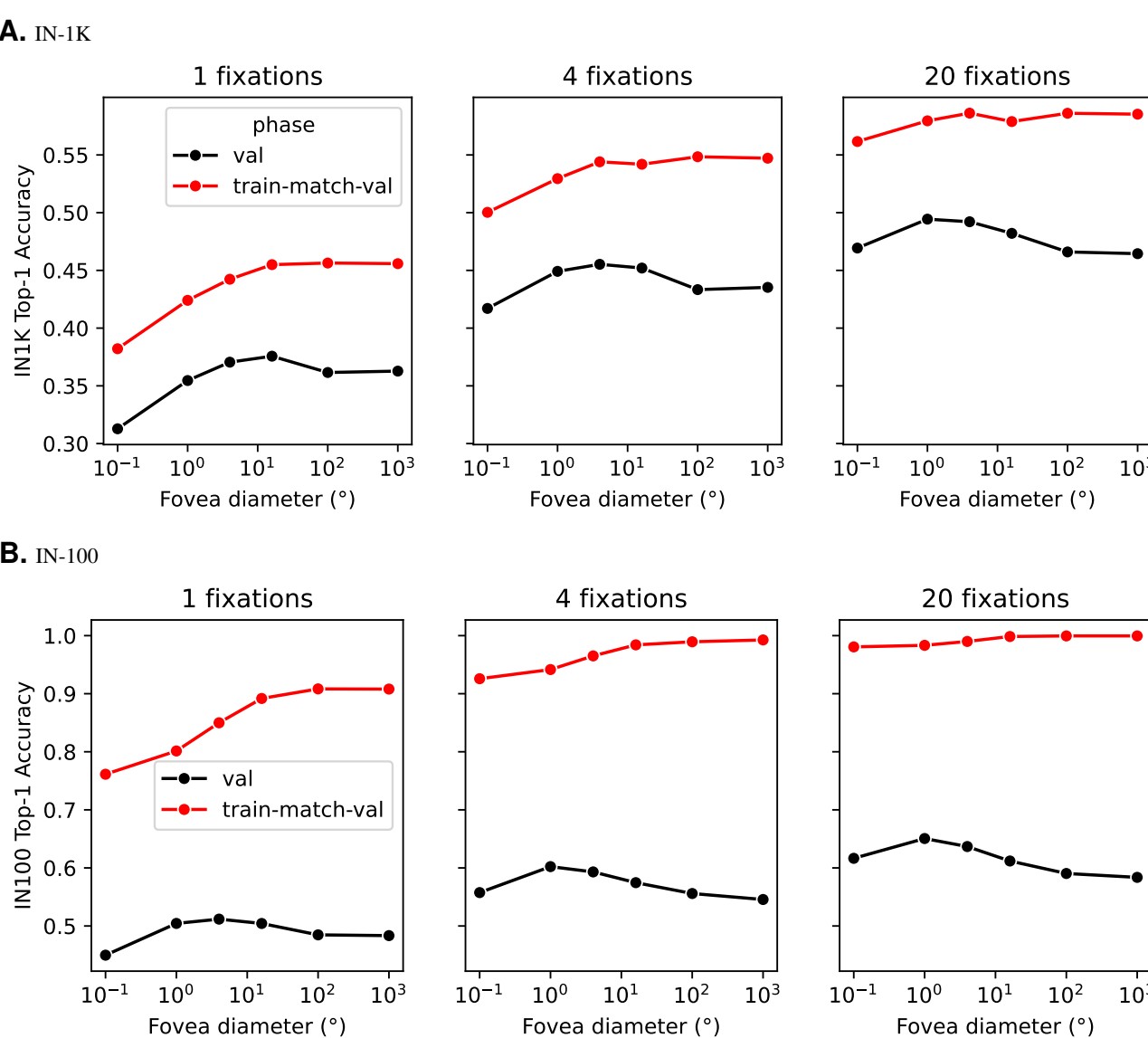

**B.** IN-100

**Figure S5.** Comparing performance for IN-1K and IN-100. For both **A.** and **B.**, we plot performance as a function of foveation parameter $a$, when evaluated on either validation images, or a matched size sample of training images evaluated without data augmentation. Columns show evaluations using either 1, 4, or 20 fixations, using standard mean-logit aggregation across fixations. **A.** IN-1K results. **B.** IN-100 results.

We next explored the hypothesis that increasing performance for intermediate foveation reflected an **optimal sampling resolution effect**: strong-intermediate foveation works best because it samples near the native resolution in the fovea, neither more (oversampling), nor less (undersampling). One prediction of this account is that decreasing the native resolution of the incoming signal, while keeping the sampling resolution the same, should shift the accuracy x $a$ curves downward (due to less effective resolution), but more importantly, rightward due to a better match of more weakly foveated models (models with larger $a$ parameter). Thus, we trained a set of matched models using images that were first resampled at a resolution of 64x64, which we call the native resolution. In Figure S6, we find that both predictions are validated, with performance decreasing overall, but more for more heavily foveated models (i.e., $a = 0.05$ vs. $a = 500$). However, the advantage for weakly foveated models remains, suggesting that the optimal sampling resolution effect does not fully explain the observed pattern of results.

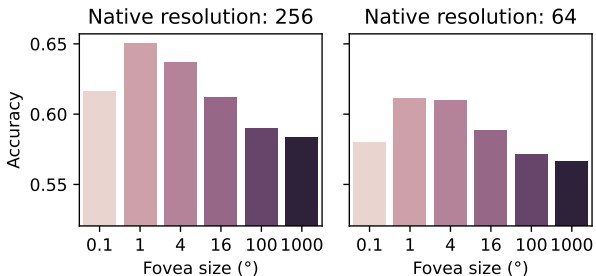 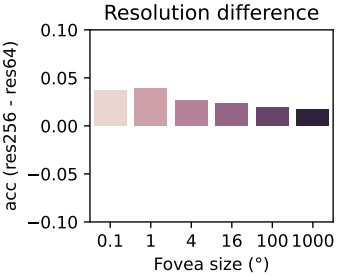

**Figure S6.** Assessing the effect of native image resolution on foveated recognition performance. In the main experiments, we use a native resolution of 256. Here, we plot accuracy using a native resolution of 64 (left) or 256 (middle; standard). On right, we take the difference of the results, for each foveation level.

Our next hypothesis was that the foveation advantage reflects a **central bias of relevant content**, where models with uniform sampling struggle to focus on the central information compared to foveated models. To better understand this, we tested models that incorporate a different form of foveation that could allow for better focus on central information: smaller crops. In our main experiments, we use full-size image crops, but here, we compare with models trained and evaluated using smaller crops of a fraction of 0.2 of the image area; for a 256x256 image, this corresponds to a 114x114 crop. Results are shown in Figure S7. We find that smaller crops lead to worse performance for a single fixation, whereas for the maximum 20 fixations, performance is identical for foveated models, but enhanced for models with more uniform sampling. This suggests that the more uniform models are able to benefit from foveation through cropping, which provides enhanced accuracy over many fixations. However, a large field-of-view provides many real world benefits; here, we capture one of such benefits, which is better performance for a smaller number of fixations.

Last, we explore sampling with a larger fixation zone, using a radius of 0.45 in place of the original radius of 0.25. Interestingly, for large crops, we actually find improved performance for a larger fixation zone, at odds with predictions of the central bias hypothesis.

**A.** 1 fixation

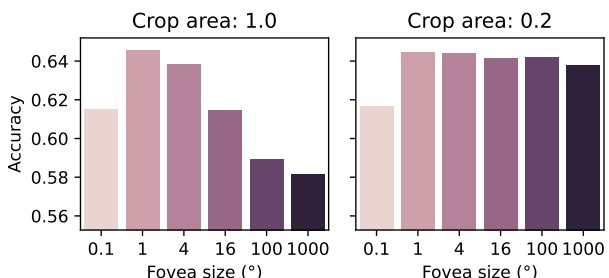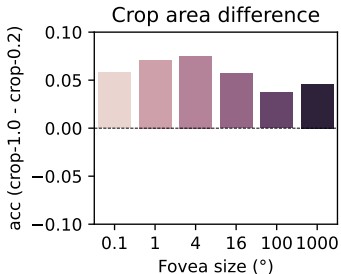

**B.** 20 fixations

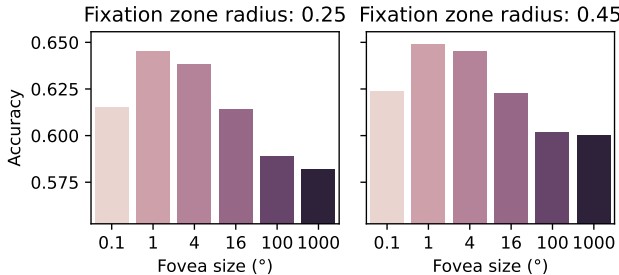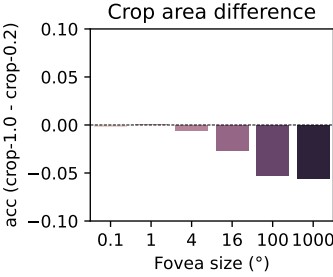

**Figure S7.** Assessing the effect of fixation crop area on foveated recognition performance. In the main experiment, we set the crop area to the full image size. Here, we additionally test a reduced fraction of the image area (0.2). We plot accuracy using the crop area of 0.2 (left) or 1.0 (middle; standard). On right, we take the difference of the results, for each foveation level.

**A.** Crop area = 1.0

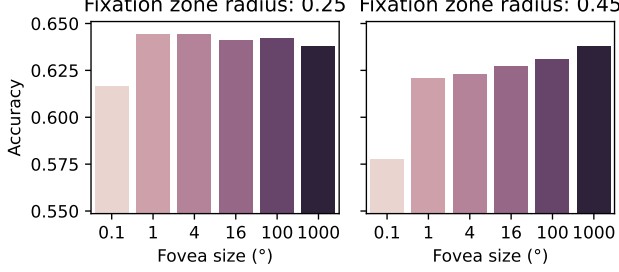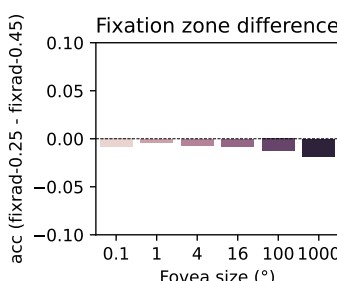

**B.** Crop area = 0.2

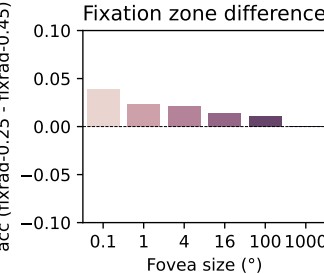

**Figure S8.** Assessing the effect of fixation zone size on foveated recognition performance. In the main experiment, we use a circular fixation zone with a radius of 0.25 of the total image diameter. Here, we compare models trained using a larger fixation zone parameterized by a radius of 0.45. In **A.**, we assess models trained with the standard large crop area (1.0). In **B.**, we assess models trained with the smaller crop area (0.2). In each subpanel, we plot accuracy using the larger fixation zone of 0.45 (left) and standard fixation zone of 0.25 (middle). On right, we take the difference of the results, for each foveation level.

## 10.5. Higher resolution reference filters improve FOVI-CNN performance

Here, we plot results for a FOVI-CNN comparing 1x and 2x filter resolution. 1x filter resolution indicates that the reference kernel has exactly k entries, and thus, the side length is $\sqrt{k}$. 2x filter resolution has side length of $2\sqrt{k}$. While the 2x filter resolution has more parameters, they are highly mutually constrained due to the fixed mappings into individual KNNs, and thus live in a lower dimensional subspace. However, this finer resolution appears to allow for better mapping into individual kernels.

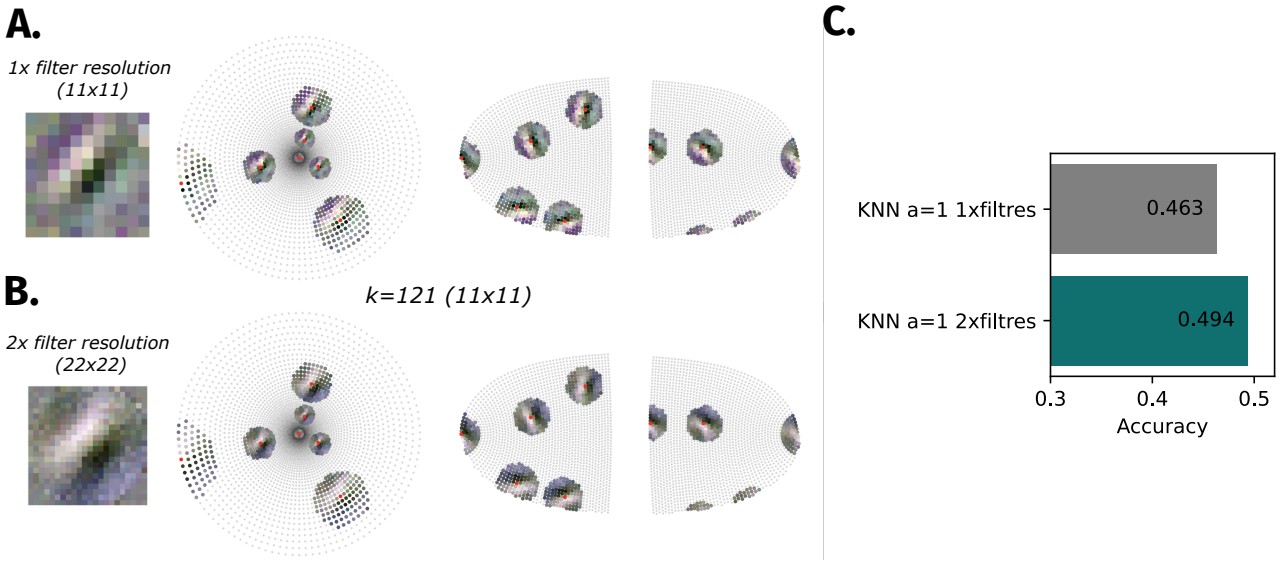

**Figure S9.** Higher reference filter resolution improves performance. **A.** Standard filter resolution for a $k = 121$ kernel (11x11). **B.** Double filter resolution for a $k = 121$ kernel (22x22). **C.** ImageNet-1k performance for foveated FOVI-CNN models with 2x and 1x filter resolution, along with a matched CNN.

## 10.6. Adapting DINOv3

We explore a variety of methods for adapting DINOv3 to take foveated inputs, comparing to a frozen baseline. Here, to facilitate many experiments, we use the IN-100 dataset as in our hyperparameter explorations of FOVI-CNN models. We first note that the frozen baseline performance is significantly reduced relative to the off-the-shelf non-foveated variant operating at a typical 224x224 resolution with 16x16 patch size (93%). Next, we finetune DINOv3 end-to-end, including the foveated patch embeddings. Relative to a frozen backbone, this leads to a significant improvement in validation accuracy, but also a large degree of overfitting (Figure S10A, top). Next, we explore fine-tuning only the first half of the ViT (first 6 layers, indexed as 0-5), along with the patch embedding. This performs similarly, albeit somewhat worse than full fine-tuning. Next, we explore low-rank adaptation (Hu et al., 2021), a method for adapting weight matrices using two low-rank matrices that has been widely successful in preventing overfitting when fine-tuning models on smaller datasets than the original pre-training dataset. LoRA uses the following equation to re-parameterize weight matrices $W$ into low-rank adaptable matrices $\hat{W}$, using two low rank matrices $A$ and $B$:

$$\hat{W} = W + \frac{\alpha}{r} * (BA) \tag{1}$$

Where $\hat{W}$ and $W$ are of shape $(d_{out}, d_{in})$, $A$ is of shape $(r, d_{in})$ and B is of shape $(d_{out}, r)$. We set $r = 8$ and $r = \alpha$ unless otherwise specified. We adapt all weight matrices within a given transformer layer, and explore adapting different combinations of layers.

We find that LoRA over the first half of the network, along with the patch embedding, leads to a significant improvement in validation accuracy (Figure S10A, bottom), along with a large reduction in overfitting. We find that adapting the first half of the network performs best of other strategies we test, including adapting the whole network, only earlier layers, and only the latter half of the network (Figure S10B).

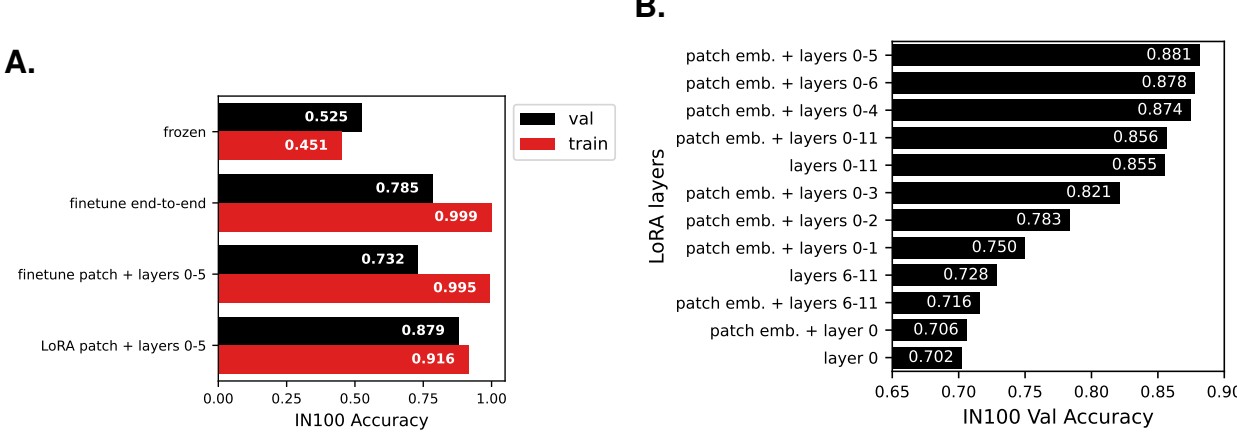

**Figure S10.** Results adapting DINOv3 to process a foveated tokenization ($a = 0.5$) on ImageNet-100, using different strategies. **A.** Comparing end-to-end finetuning, with early layer finetuning, and early-layer LoRA, for train and val accuracy. Early-layer LoRA training leads to superior generalization and less overfitting. **B.** Comparing a range of LoRA recipes, varying which layers are subject to adaptation. Adapting the patch embedding and first 6 layers produces the best performance in this setting, with significantly worse performance adapting only the first layer or only the late layers.

### 10.7. Performance as a function of foveation degree

Next, we examine the dependence of performance on the foveation parameter $a$. Due to the small number of patches, the inability to exactly determine the number of samples at a given foveation parameter ($a$) can lead to a large percent difference in patch count across models. Thus, rather than choose $a$ directly, here, we constrain the set of $a$ values to those which produce exactly the desired number of patches. For $n = 64$ patches, we determine 4 suitable $a$ values (rounded here to two decimal places): (0.03, 0.14, 0.58, 2.79, 60.94), as shown in Figure S11.

**A.** **B.**

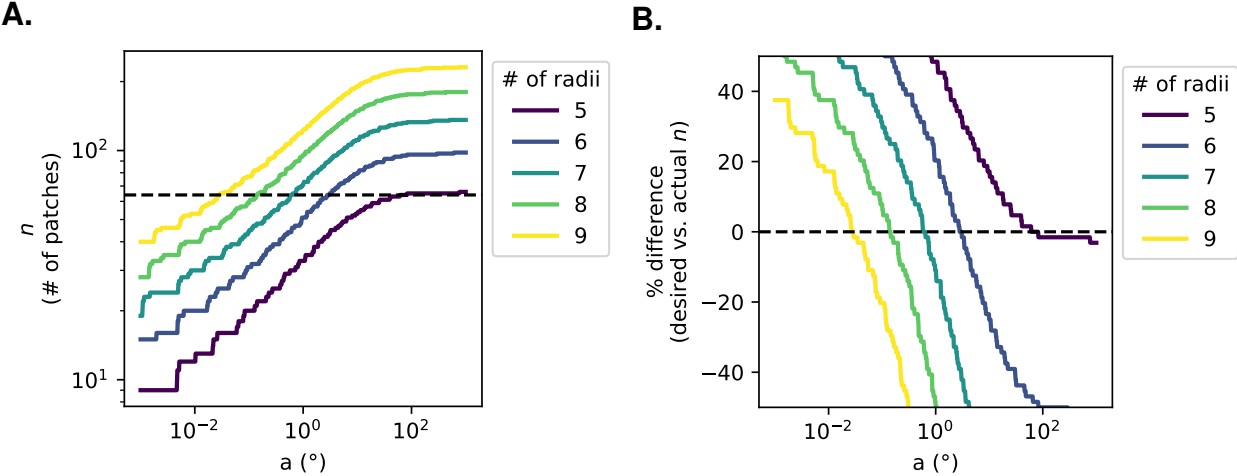

**Figure S11.** Determining a set of $a$ values that produce the exact desired number of patches. **A.** Resulting $n$ (here, number of patches) across different combinations of foveation parameter ($a$) and # of sampling radii; note: the # of radii is specified, and the number of sampling points is determined in order to satisfy local isotropy given the particular $a$ value, so it is not fully controllable. **B.** Percent difference in produced $n$ vs. actual $n$. Here, since $n$ is small, the percent difference can be very large. However, by finding the intercepts of each curve, we can specify a set of $a$ values that satisfy a perfect match to the desired $n$. Note: we specify a single $a$ per model, so the number of pixels is not exactly matched across models, however the percent difference is much smaller since the desired $n$ is much larger (4096 in our main experiments); for the pixel sampling array, we set the # of radii to the value that most closely matches the target $n$, while not exceeding it.

We plot the results in Figure S12, using IN-100. We find smaller effects here than with the AlexNet-like CNNs, and a peak performance using $a = 2.79$ rather than $a = 0.58$. However, we see a similar inverted U-shaped curve, suggesting again that an intermediate level of foveation is ideal.

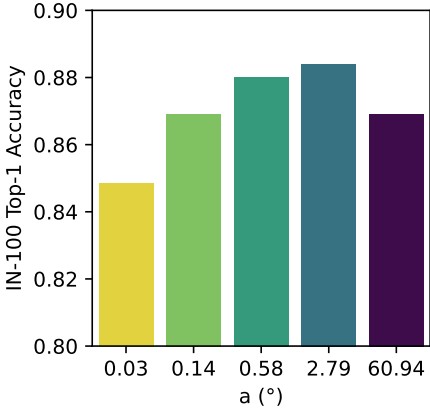

**Figure S12.** Foveated DINOv3 performance on IN-100 as a function of the foveation parameter $a$.

## 10.8. Extended DINOv3 analyses

In the main paper (Table 1), we focused on baseline uniform @ 224 models, and reduced patch/pixel models using approximately 4096 pixels and 64 patches. Here, we include some other models trained with different pixel counts and the same number of patches. We find that higher resolution helps, even with the same number of patches, since the patches can integrate finer-grained information.

| | # fix. | pixels | patches | GFLOPS | acc. [val] | lat. (ms) [train] | lat. (ms) [val] | mem. (GB) [train] | mem. (GB) [val] |
|---|---|---|---|---|---|---|---|---|---|
| ViT-H+ uniform (224 x 1-fix) | 1 | 50176 | 196 | 172.39 | 0.871 | 289.59 | 103.80 | 40.1 | 4.1 |
| FOVI-ViT-H+ @ 64 (a=2.79) | 1 | 3976 | 64 | 58.43 | 0.844 | 119.95 | 45.01 | 19.1 | 4.1 |
| FOVI-ViT-H+ @ 64 (a=2.79) | 3 | 11928 | 192 | 175.30 | 0.853 | 303.70 | 111.76 | 40.9 | 4.2 |
| ViT-S+ uniform @ 224 | 1 | 50176 | 196 | 6.16 | 0.794 | 37.92 | 15.53 | 4.1 | 1.0 |
| FOVI-ViT-S+ @ 224 (a=2.79) | 1 | 50104 | 64 | 2.10 | 0.737 | 27.66 | 10.76 | 1.9 | 1.2 |
| Weak FOVI-ViT-S+ @ 224 (a=60.94) | 1 | 49860 | 64 | 2.11 | 0.734 | 27.77 | 10.81 | 1.9 | 1.2 |
| FOVI-ViT-S+ @ 128 (a=2.79) | 1 | 16078 | 64 | 2.05 | 0.729 | 27.82 | 10.67 | 1.8 | 1.1 |
| FOVI-ViT-S+ @ 64 (a=2.79) | 1 | 3976 | 64 | 2.04 | 0.700 | 27.61 | 10.84 | 1.7 | 1.0 |
| ViT-S+ uniform @ 64 | 1 | 4096 | 64 | 2.02 | 0.693 | 27.60 | 10.79 | 1.7 | 1.0 |
| Weak FOVI-ViT-S+ @ 64 (a=60.94) | 1 | 4032 | 64 | 2.04 | 0.693 | 27.41 | 10.76 | 1.7 | 1.1 |
| ViT-S+ log-polar @ 64 (a=2.79) | 1 | 4096 | 64 | 2.02 | 0.643 | 27.86 | 10.77 | 1.7 | 1.0 |
| FOVI-ViT-S+ @ 224 (a=2.79) | 3 | 150312 | 192 | 6.30 | 0.765 | 49.00 | 25.93 | 4.6 | 1.3 |
| FOVI-ViT-S+ @ 128 (a=2.79) | 3 | 48234 | 192 | 6.16 | 0.760 | 46.89 | 23.69 | 4.3 | 1.1 |
| Weak FOVI-ViT-S+ @ 224 (a=60.94) | 3 | 149580 | 192 | 6.32 | 0.756 | 48.32 | 25.84 | 4.6 | 1.3 |
| FOVI-ViT-S+ @ 64 (a=2.79) | 3 | 11928 | 192 | 6.12 | 0.735 | 46.78 | 23.63 | 4.3 | 1.1 |
| ViT-S+ uniform @ 64 | 3 | 12288 | 192 | 6.06 | 0.726 | 46.66 | 23.47 | 4.2 | 1.1 |
| Weak FOVI-ViT-S+ @ 64 (a=60.94) | 3 | 12096 | 192 | 6.12 | 0.725 | 46.76 | 23.59 | 4.3 | 1.1 |
| ViT-S+ log-polar @ 64 (a=2.79) | 3 | 12288 | 192 | 6.06 | 0.694 | 46.89 | 23.54 | 4.2 | 1.1 |

**Table 2.** Performance comparison of DINOv3 variants, adding in @ 128 and @ 224 variants not included in Table 1. The table is divided into H+ and S+ variants, then sorted by number of fixations, and then sorted by accuracy. GFLOPs are reported per image, while latency and memory are computed in both training and validation modes with a batch size of 64, using a single NVIDIA H100 GPU. In training mode, both forward and backwards passes are run; in validation mode only a forward pass is run, with gradients disabled. "@ 64" indicates using a sampling resolution parameter of 64, equivalent to approximately $64^2$ pixels, and $a$ refers to the foveation parameter (smaller = stronger foveation). Uniform full-resolution baselines are run only for 1 fixation. Abbreviations: accuracy (acc.), fixation (fix.), latency (lat.), peak memory (mem.).

