# OpenReview forum: "FOVI: A biologically-inspired foveated interface for deep vision models"
_ICML.cc/2026/Conference — ICML 2026 regular_

### Official Review · Reviewer_HDDR · 2026-03-05

**Soundness:** 3
**Presentation:** 3
**Significance:** 2
**Originality:** 2
**Overall Recommendation:** 4
**Confidence:** 4

**Summary:**

The paper develops a biologically-inspired way of sensing and processing visual input, which allocates higher-resolution and more compute towards pixels around the fixation point (fovea). For transformer architectures, this simply results in a different tokenization technique. The paper demonstrates the usefulness of foveation compared to uniform sampling in resource-constrained regimes.

**Compliance With Llm Reviewing Policy:**

Affirmed.

**Final Justification:**

I thank the authors for their rebuttal. I still believe the empirical evaluation is rather weak. The major root of this concern, however, is rather in the presentation of the existing results. Even w/o having clear gains, i am aligned with other reviewers on the value of the work in advancing the subarea of foveated vision, and that it can be meaningfully built upon. So, i am inclined to recommend the acceptance under the following (realistic) revision:

- The authors should present findings in Tab. 1 more clearly, so the trends w.r.t. both flops and pixels are clear to the reader. Specifically, it should be clear that 1) in "low-flops" (2 GFLOPs) the method is on par with uniform (maybe, slightly outperforming) and 2) in "larger-flops" (6 GFLOPs) it lags behind it. I would also suggest densifying it and presenting as a figure, instead.
- Adjust wording to not overclaim the existing results, specific examples:
  - Abstract, L029: "These models provide competitive performance at a fraction of the computational cost of non-foveated baselines" -     - I still do not see strong evidence for this.
  - From authors' final response: "our method outperforms uniform sensing at the matched reduced FLOP setting at both @ 64 and @ 224 resolution" -- "outperforms" is rather strong here, it's 0.737 vs. 0.734 (which is also "weak FOVI", not actual "Uniform", which the authors should also include).

**Key Questions For Authors:**

- How does this method compare to other inference-cost reduction techniques in ViTs (including using larger patches)? What is the main practical advantage of FOVI compared to those methods?
- What are the implications of FOVI for other vision tasks, especially dense ones? Can it perform them well?
- Comparing `FOVI-ViT-S+ @ 64 (a = 2.79) = 0.700` vs. `Weak FOVI-ViT-S+ @ 64 (a = 60.94) = 0.693` suggests that the improvement is mainly not due to the foveation but another component of the method. Did the authors further ablate the method to identify the main cause? How does it stack against the overall conclusion of the benefits of foveation?
- Why do authors include only a single "fovea-inspired" baseline, while the related work mentions a few other methods?

**Limitations:**

The paper includes a "limitations and future research" section, but does not clearly state the method's or evaluation's actual limitations. Please see the weaknesses for the relevant limitations of the work.

**Strengths And Weaknesses:**

**Strengths:**
- The paper provides a relatively unique biologically-inspired perspective on image tokenization.
- The proposed method is well-motivated and addresses some limitations of the previous methods. It appears relatively simple, especially for ViT architectures, and can be useful for the community as a building block for further research.
- The paper is overall well-written, providing useful discussions and visualizations.

**Weaknesses:**
While theoretically sound, my major concerns are the overall limited evaluations and practical utility.

- No clear performance-versus-inference-cost efficiency trends have been established. ViT is a good testbed for this, as the proposed method only changes the tokenization step, and Tab. 1 already provides a glimpse into this trend. Including performance vs. FLOPs across various FLOPs levels would make the evaluation more comprehensive and clearer.
	- For example, comparing same-FLOPS FOVI-ViT-S+ @ 64 @ 3 fixations 0.760 vs. ViT-S+ Uniform @ 224 0.794 suggests that the proposed scaling axis (#fixations) of the FOVI ViT lags behind the standard ViT tokenization, suggesting no simple swap. Same for ViT-H.
	- From the tokenization and FLOPs perspective, I believe that Uniform @ 64 should be paired with a potentially stronger baseline of using larger patches, resulting in 64 total tokens, as it's the number of tokens that mainly governs the FLOPs, not the pixel count.
	- Finally, the discussion in Section 6.3 is not supported by experiments measuring performance at different resolutions.
- The foveation evaluations in Fig. 4 use a relatively old AlexNet architecture. It remains unclear whether similar trends hold for larger networks, e.g., ResNets.
- The study focuses on a single ImageNet classification task. It is unclear what the implications of this approach are for other vision tasks, such as detection, segmentation, and captioning.
- Given that for ViTs the main change seems to be the patchification technique, comparison to other techniques that reduce the number of tokens or interactions between them would be needed to better position this work among relevant techniques, e.g.:

[1] Liu, Ze, et al. "Swin transformer: Hierarchical vision transformer using shifted windows.", ICCV 2021

[2] Wang, Wenhai, et al. "Pyramid vision transformer: A versatile backbone for dense prediction without convolutions.",  ICCV 2021

[3] Wang, Feng, et al. "Scaling laws in patchification: An image is worth 50,176 tokens and more.", arXiv 2025

[4] Fan, Haoqi, et al. "Multiscale vision transformers.", ICCV 2021


More minor points:
- The conclusion highlights SoTA results in the "low-pixels" regime, which is not the real advantage, as the number of patches (tokens) determines the actual FLOPs (as seen with the @64 and @128 settings in Tab. 1).
- How does the cost of the custom kNN patchification layer compare to standard uniform squared patchification? What is its contribution to the overall inference cost? This discussion should be included in the paper. (I assume it's ~0.02 based on Tab. 1, which seems moderate, but is worth mentioning)

---

> ### Author Rebuttal · Authors · 2026-03-31
>
> Thank you for the thoughtful and detailed feedback. We appreciate the recognition that the paper offers a “relatively unique biologically-inspired perspective on image tokenization,” is “well-motivated,” and may be “useful for the community as a building block for further research.” First, we ask that you see the short "general reply to all reviewers" in our response to Reviewer F27X above.
>
> As the weaknesses and questions are largely overlapping, we provide general bulleted responses to the main points raised across both:
> - On efficiency trends and equal-compute comparisons. We agree that more explicit performance-versus-cost curves would be useful. Our intent was for this to be conveyed jointly by Table 1 and Figure 5: Table 1 reports concrete operating points with accuracy, latency, pixels, patches, and GFLOPs, while Figure 5 analyzes how cost scales with image resolution and fixation count in the ViT setting. If allowed, we would be happy to add a larger sweep over models in a figure as recommended by the reviewer, however, we do not believe it is necessary to convey the point that our foveated models provide competitive performance at reduced computational cost.
> - We have added a new set of 224-res, 64-patch FOVI and weak FOVI models, as recommended by the reviewer, which use larger patches rather than downsampling, along with variable foveation strength in their resampled pixel distribution. The FOVI model still slightly outperforms the weak FOVI baseline (0.737 vs. 0.734 for 1 fixation, 0.765 vs. 0.756 for 3 fixations), and these new models demonstrate that foveated re-sampling (rather than just downsampling) can be useful when pixel sampling costs are minimal.
> - On the role of the CNN experiments. We appreciate the concern that the CNN experiments use an AlexNet-like architecture. The intended purpose of that section is primarily mechanistic, not to serve as the strongest modern benchmark. The CNN section demonstrates that the proposed interface supports end-to-end hierarchical learning directly on the manifold, and that it reproduces biologically relevant receptive-field properties, including eccentricity-dependent receptive-field scaling and shape. The more practically oriented benchmark is then the DINOv3 ViT section, where the same interface is integrated into a strong contemporary foundation model. The CNN results are meant to establish that the interface is not merely a special tokenization trick for ViTs, but a general processing substrate that can support hierarchical feature learning within deep vision models more generally.
> - On broader tasks such as detection, segmentation, and captioning. We agree that broader downstream evaluations are important. The current submission should be read as establishing the sensor/interface and processing mechanism, rather than as exhausting all downstream applications. We will acknowledge this limitation.
> - We agree that other token reduction and efficiency methods are highly relevant neighboring work, and we will add further references to better situate our work in the camera-ready version. As we state in our general response, we believe that the added pixel reduction provides additional benefits in many possible extensions of our work, and is in line with the broad prior interest in foveated sensing approaches.
> - The cost of kNN patchification is extremely minimal. After correcting for incompletely incorporated RoPE embeddings in the non-FOVI models, there is essentially no difference in latency (<0.1%), and 1% difference in overall FLOPS.
> - On whether the gains are really due to foveation. We appreciate the question. As discussed in the general response, we found that the difference between weak FOVI and uniform models – which should in principle yield highly similar performance – arose due to an implementation bug in adapting the RoPE module of non-FOVI models to different image / patch resolutions. We’ve fixed this and the results are now comparable. More generally, the paper’s claim is not that “foveation always wins,” but that under constrained sensing there is a useful intermediate regime where non-uniform allocation improves the utility of a limited sample budget. We observe this pattern in both the CNN and ViT settings: in the CNN experiments, performance peaks at intermediate foveation levels, and in the ViT appendix there is again an inverted-U relationship between performance and foveation strength. Note that the performance gap increases with more fixations, both for the CNN (Figure 3D) and ViTs (Table 1).
>
> We appreciate the reviewer’s feedback, and we hope our responses make the intended contribution and scope of the paper more precise.

---

> > ### Author Rebuttal · Reviewer_HDDR · 2026-04-03
> >
> > I thank the authors for their response. I would like to reiterate that i believe that this direction is interesting and should be explored by the community. The proposed method is simple and sound. However, i think, the empirical evaluation is too weak for acceptance in the (largely) empirically driven field. The overall evaluation is rather narrow (tasks, architectures, inference compute regimes), and the existing results do not provide clear evidence that the FOVI method is beneficial when accounting for compute to warrant adoption (see below).
> >
> > **On the efficiency:**
> > The paper and the response still do not concretely discuss the value of reducing the number of pixels *at the same inference FLOPs cost*. When comparing at different FLOPs budgets, there is no evidence for the FOVI models to be better than standard Uniform patchification. It appears that at ~6.X GFLOPs, there is no advantage of FOVI as the Uniform@224 performs the best. There seems to be an advantage at 2.X GFLOPs, but Uniform @ 224 with 64 patches baseline is missing to make this conclusion. Below i attached slightly different view of Tab. 1 to make it more clear.
> >
> > ### ~6 GFLOPs
> >
> > | Model          | Resolution | Patches | Fixations | Accuracy  | Latency (ms) | GFLOPs |
> > | -------------- | ---------- | ------- | --------- | --------- | ------------ | ------ |
> > | ViT-S+ Uniform | 224        | 196     | 1         | **0.794** | 50.18        | 6.16   |
> > | FOVI-ViT-S+    | 224        | 64      | 3         | 0.765     | --           | --     |
> > | FOVI-ViT-S+    | 128        | 64      | 3         | 0.760     | 76.10        | 6.16   |
> > | FOVI-ViT-S+    | 64         | 64      | 3         | 0.735     | 74.04        | 6.11   |
> > | ViT-S+ Uniform | 64         | 64      | 3         | 0.726     | 68.07        | 6.05   |
> >
> > ### ~2 GFLOPs
> >
> > | Model                  | Resolution | Patches | Fixations | Accuracy | Latency (ms) | GFLOPs   |
> > | ---------------------- | ---------- | ------- | --------- | -------- | ------------ | -------- |
> > | *ViT-S+ Uniform @ 224* | *224*      | *64*    | *??*      | *??*     | *??*         | *~2 (?)* |
> > | FOVI-ViT-S+            | 224        | 64      | 1         | 0.737    | --           | --       |
> > | FOVI-ViT-S+            | 128        | 64      | 1         | 0.729    | 26.43        | 2.05     |
> > | FOVI-ViT-S+            | 64         | 64      | 1         | 0.700    | 26.36        | 2.04     |
> > | ViT-S+ Uniform         | 64         | 64      | 1         | 0.693    | 24.50        | 2.02     |
> >
> > **Additional comments**
> >
> > > FOVI is intentionally a sensing interface
> >
> > The paper does not discuss, however, the implications of the proposed method on the actual sensors and builds upon an assumption of having a standard camera sensor.
> >
> >
> >
> > >We observe this pattern in both the CNN and ViT settings: in the CNN experiments, performance peaks at intermediate foveation levels, and in the ViT appendix there is again an inverted-U relationship between performance and foveation strength.
> >
> > This is an interesting and valid observation. My main practical concern is mainly about this peak performance lagging behind the standard tokenization at the matched inference flops budget (see above).

---

> > > ### Author Response · Authors · 2026-04-08
> > >
> > > Thank you for the continued engagement with our paper. We do appreciate your noting that our method for foveated perception is simple and sound, and that the work is of strong relevance to the community. We are pleased that the other three reviewers agree and have recommended acceptance.
> > >
> > > ### **The main remaining concern**:
> > > "Peak performance lagging behind the standard tokenization at the matched inference flops budget".
> > >
> > > This is a valid concern, however, we do want to reiterate that our method outperforms uniform sensing at the matched reduced FLOP setting at both @ 64 and @ 224 resolution, 64 patch settings (note that after our update, "weak fovi" can serve as an approximate stand in for uniform). Beyond this, our model significantly reduces pixel count to the base full resolution model, and provides an alternative low FLOP setting that retains competitive performance. Here is a filled in version of the table you shared:
> > >
> > > **~6 GFLOPs**
> > > | Model          | Resolution | Patches | Fixations | Accuracy  | Latency (ms) | GFLOPs | Pixels |
> > > | -------------- | ---------- | ------- | --------- | --------- | ------------ | ------ | ------ |
> > > | ViT-S+ Uniform | 224        | 196     | 1         | 0.794 | 50.18        | 6.16   | 50176  |
> > > | FOVI-ViT-S+    | 224        | 64      | 3         | 0.765     | 83.8         | 6.29   | 150312 |
> > > | FOVI-ViT-S+    | 128        | 64      | 3         | 0.760     | 76.10        | 6.16   | 48234  |
> > > | FOVI-ViT-S+    | 64         | 64      | 3         | 0.735     | 74.04        | 6.11   | **11928**  |
> > > | ViT-S+ Uniform | 64         | 64      | 3         | 0.726     | 68.07        | 6.05   | 12288  |
> > >
> > > **~2 GFLOPs**
> > > | Model                                  | Resolution | Patches | Fixations | Accuracy | Latency (ms) | GFLOPs | Pixels |
> > > | -------------------------------------- | ---------- | ------- | --------- | -------- | ------------ | ------ | ------ |
> > > | FOVI-ViT-S+                            | 224        | 64      | 1         | 0.737    | 28.88        | 2.10   | 50104  |
> > > | Weak Fovi-ViT-S+ (approx uniform)      | 224        | 64      | 1         | 0.734    | 28.94        | 2.11   | 49860  |
> > > | FOVI-ViT-S+                            | 128        | 64      | 1         | 0.729    | 26.43        | 2.05   |  16078     |
> > > | FOVI-ViT-S+                            | 64         | 64      | 1         | 0.700    | 26.36        | 2.04   | **3976**   |
> > > | ViT-S+ Uniform                         | 64         | 64      | 1         | 0.693    | 24.50        | 2.02   |   4096     |
> > >
> > > Additionally, as shown in our scaling analyses, the efficiency gains grow significantly with higher resolution, where the quadratic attention cost begins to dominate against the linear costs of MLPs which dominate in the currently tested regime.
> > >
> > > ### **The auxiliary point**:
> > > "The paper does not discuss, however, the implications of the proposed method on the actual sensors and builds upon an assumption of having a standard camera sensor."
> > >
> > > We are glad to address this here and in the camera-ready paper. FOVI makes no assumption of having a standard camera sensor. The current setting is a pixel sampling setting. However, as we noted in the paper, the same sampling grid can be used to directly control ray generation in graphics rendering settings, using a simulated camera. Long term, we believe foveated camera design is an intriguing area of possible development because it could allow for a very high peak-resolution camera at very low power compared to a uniform camera of similar peak resolution, and this has been explored for log-polar and other forms of foveated sensor arrays. FOVI is at its heart a simple and precise mathematical framework linking a locally-isotropic eccentricity-dependent sampling array with a manifold on which regular (i.e. convolutional) processing can be performed, along with a method for performing that processing (kNN convolution), and a demonstration of its success in modern deep learning (using CNNs and ViTs). Whether the FOVI "retinal" sensor layout is used for sampling pixels an image, generating rays in rendering, or sampling photons in a sensor, is determined by the specific use case.

---

### Official Review · Reviewer_tRtQ · 2026-03-08

**Soundness:** 3
**Presentation:** 3
**Significance:** 3
**Originality:** 3
**Overall Recommendation:** 4
**Confidence:** 2

**Summary:**

The paper introduces FOVI, a biologically inspired foveated vision interface designed to enhance efficiency in deep vision models by mimicking the human visual system's variable-resolution sampling, which peaks at the center of gaze and decreases peripherally. Drawing from models of the primate retina and primary visual cortex (V1), FOVI reformats a non-uniform sensor array in visual space into a uniformly dense sensor manifold, enabling regular processing through a novel k-nearest-neighbor (kNN) convolution operation facilitated by a kernel mapping technique that ensures aligned and isotropic receptive fields. The authors demonstrate two primary applications: (1) an end-to-end foveated convolutional neural network (FOVI-CNN) that learns hierarchical features over the manifold, showing biologically plausible receptive field properties and improved ImageNet classification performance at intermediate foveation levels under pixel constraints; and (2) a foveated adaptation of pre-trained DINOv3 Vision Transformers (FOVI-ViTs) using low-rank adaptation (LoRA) for patch embedding, achieving competitive accuracy with reduced computational costs compared to uniform-resolution baselines. Overall, FOVI provides a general-purpose interface for scalable, high-resolution active sensing in applications like robotics and egocentric vision.

**Compliance With Llm Reviewing Policy:**

Affirmed.

**Key Questions For Authors:**

1. Could you provide more details on how the kernel mapping handles edge cases, such as highly eccentric patches where kNNs might include significant padding? A response clarifying robustness (e.g., via ablation studies) could strengthen my confidence in the method's soundness.
2. In the FOVI-ViT experiments, how sensitive are results to the choice of LoRA rank and the specific pre-trained DINOv3 variants? Providing quantitative sensitivity analysis could clarify the method's generalizability.

**Limitations:**

yes

**Strengths And Weaknesses:**

**Strengths:**

1. The paper is technically sound, with claims well-supported by a combination of theoretical derivations, simulations, and empirical experiments. The proposed foveated sampling and kNN-convolution are grounded in established biological models (e.g., Rovamo & Virsu, 1984; Schwartz, 1980), and the assumptions, such as using the cortical magnification function for isotropic sampling, are reasonable and explicitly justified.
2. The submission is clearly written and well-structured, with a logical flow from biological motivation to technical details, contributions, and evaluations.
3. The work offers new insights by creatively combining biological models of retino-cortical mapping with modern deep learning techniques, introducing a novel foveated interface that supports end-to-end learning without the anisotropy issues of prior methods.

**Weaknesses:**

1. Although GFLOPs are significantly lower, the actual inference time (Latency) of kNN searches and irregular memory accesses tends to be higher than that of highly optimized matrix multiplications (standard convolutions) on standard GPUs. Although Table 1 in the article mentions Latency, the discussion of hardware acceleration optimization in complex scenarios is still insufficient.
2. Table 1 shows latency, GFLOPs, pixels and patches, but some comparisons still mix multiple factors: for example, the high-resolution uniform baseline uses a single fixation, while the FOVI model can use multiple fixations.

---

> ### Author Rebuttal · Authors · 2026-03-31
>
> Thank you for the constructive and detailed feedback, and for recognizing the technical grounding, clarity, and originality of the work. Responses to weaknesses:
>
> 1. We agree that irregular operations can weaken the correspondence between GFLOPs and wall-clock speed on standard GPUs. As the reviewer notes, this is why we reported latency in Table 1 in addition to GFLOPs, rather than relying on FLOPs alone. We will add further detail to this important topic. Indeed, there are multiple possible implementations of the kNN convolution which expose computational trade-offs under different settings. In the open-source code-base that we plan to release as a companion to this paper, we implement multiple kNN convolution backends and allow the user to select the one which is most efficient under their particular use-case. Additionally, the overhead of kNN operations is very minimal for ViTs which have only a single large-stride kNN convolution. For our ViT-H+ architecture with 1 fixation, we achieve 96.4% of the accuracy at 38.4% of the latency, 33.9% of the FLOPS, and 8.1% of the pixels. Since we did not do an exhaustive architecture search over foveated patchification schemes, potential gains could be even larger in the limit.
>
> 2. The intent here was for the high-resolution (@224) variants to be the standard 1-fixation baseline, with the @64 variants compared under two settings: 1 fixation (resource-constrained), and 3 fixations (approximately resource-matched). More fixations of the base model would increase, rather than decrease, baseline computational costs. We hope this clarifies things.
>
> Answers to questions:
> 1. Yes - the implementation handles edge cases by extending sampling beyond the processing field-of-view and designating those locations as padding units whose activations are fixed to zero. These padding units are then naturally selected in later kNNs when neighborhoods extend beyond the valid field-of-view. We did try to highlight highly padded units specifically in Figure 3.3 (note there is a missing arrow from “unit 2” to the eccentric, padded unit in the right panel, which we will fix), and our RF analyses in Figure 4B also demonstrate the plateau due to padding. Our padding technique ensures that receptive fields do not bleed further into the central field due to a greedy kNN operation looking for non-padding units. We are happy to add a further methods diagram of this specific aspect of the method in the Appendix. We are very confident in the robustness of the technique.
>
> 2. Re: the LoRA rank, we simply selected r=8 for IN-100, and tried both r=8 and r=64 for IN-1000, with r=64 working slightly better with the larger dataset size. We swept extensively over the specific LoRA layers to adapt in Figure S10, which we think may reduce the reviewer’s concern about robustness; generally, adapting about the first half of the network worked well, with similar but slightly worse performance for adapting the whole network, both of which significantly outperformed frozen adaptation and full fine-tuning, along with other layer subsets. The appendix also reports sensitivity to the foveation parameter a, showing an inverted-U trend with intermediate foveation performing best. We do not make strong claims about the particular adaptation protocol; as in any method of machine learning, hyperparameters must be tuned to individual use cases, with original choices serving as an intuitive guide but not a definitive recipe. We are happy to include a sweep over LoRA rank as an additional supplementary figure in the camera ready. We believe the success of two DINOv3 variants (S+ and H+) also strengthens the robustness of our approach; in unreported analyses, we also found success with the standard S and H (non +) variants, which use a simpler MLP in place of the SwiGLU FFN network.
>
> Please also see the short "general reply to all reviewers" in our response to Reviewer F27X above.

---

> > ### Author Rebuttal · Reviewer_tRtQ · 2026-04-01
> >
> > The author solved my problems and I keep my score.

---

> > > ### Author Response · Authors · 2026-04-08
> > >
> > > Thanks very much for the productive review.

---

### Official Review · Reviewer_GSBi · 2026-03-09

**Soundness:** 3
**Presentation:** 3
**Significance:** 4
**Originality:** 3
**Overall Recommendation:** 5
**Confidence:** 3

**Summary:**

This paper introduces a new foveated vision interface (FOVI) for deep vision models using k-nearest neighborhoods (kNN) convolution and a kernel mapping method. The proposed approach produces biologically plausible spatial receptive fields and achieves competitive performance while requiring only a fraction of the computational cost compared to non-foveated baselines.

**Compliance With Llm Reviewing Policy:**

Affirmed.

**Final Justification:**

My concerns have been addressed during the rebuttal, and I strongly recommend this paper for acceptance.

**Key Questions For Authors:**

No

**Limitations:**

yes

**Strengths And Weaknesses:**

Strengths:
1. This paper is technically sound and well structured.
2.  This paper proposes a novel method for sampling points in a foveated manner and develops a kNN-convolution mechanism, integrating them into existing AI models to pave the way for aligning model perceptual abilities with those of humans.

Weaknesses:
1.  The models listed in Table 1 are somewhat confusing for comparison. For example, there should be a FOVI-ViT-H+ @ 224 model included to compare with ViT-H+ Uniform @ 224, so that the effect of the proposed FOVI method can be clearly evaluated.
2. There is a lack of baseline comparisons with other deep learning models for foveal vision, such as [1].

[1] Lukanov, H., König, P. and Pipa, G., 2021. Biologically inspired deep learning model for efficient foveal-peripheral vision. Frontiers in Computational Neuroscience, 15, p.746204.

---

> ### Author Rebuttal · Authors · 2026-03-31
>
> Thank you for the positive assessment and for highlighting both the technical soundness and the novelty of the proposed interface. Responses to proposed weaknesses are provided as numbered:
>
> 1. Thank you for this suggestion. We agree that the table benefits from a clearer explanation of what is being isolated in each comparison. Our main goal in Table 1 was to compare resource-constrained variants at low token/pixel budgets, alongside the base model at original resolution. A FOVI-ViT-H+@224 (with corresponding increase in number of patches) would be overkill for the current setting, since it would strongly oversample the center of gaze without reducing token count. However,  in response to another reviewer, we have added a FOVI-ViT-S+ @ 224 (64 patch) model to demonstrate the performance when decreasing patch count via larger patches on a foveated sensor manifold, rather than downsampling.
>
> 2. We appreciate the reference to another foveated vision model. Our paper’s positioning is that prior foveated sensors that re-express the input as a grid-like image—log-polar and warped-Cartesian variants included—induce local anisotropy and correspondingly warped receptive fields, whereas FOVI is explicitly designed to produce locally isotropic sampling on a cortical-like manifold. The manuscript discusses this broader family of prior work, including warped-Cartesian approaches, and explains why FOVI differs technically from them in both geometry and processing. We agree that referencing Lukanov et al. explicitly is valuable, and we will do so in the camera-ready.  This paper falls under the family of warped cartesian sensors, taking advantage of the Foveal Cartesian Geometry (FCG) method (Martinez and Robles, 2006) of selecting concentric squares of pixels that can be reshaped back into a square Cartesian image. This method thus suffers from the same issues of other warped cartesian sensors, which we describe in detail in our appendix. Fundamentally, our method is the only eccentricity-magnified sensor that will not suffer from these issues, because it is uniquely defined by the properties of 1) local isotropy, 2) eccentricity-dependent magnification and 3) global continuity; no other prior sensors share these three properties.
>
> Broadly, we appreciate your support and acknowledgement of “excellent” significance, and we will address the points raised as mentioned above. Please also see the short "general reply to all reviewers" in our response to Reviewer F27X above.

---

> > ### Author Rebuttal · Reviewer_GSBi · 2026-04-01
> >
> > Thanks for the rebuttal. I will maintain the positive score to support acceptance.

---

> > > ### Author Response · Authors · 2026-04-08
> > >
> > > Thanks very much for your helpful thoughts in reviewing the paper.

---

### Official Review · Reviewer_F27X · 2026-03-13

**Soundness:** 3
**Presentation:** 4
**Significance:** 4
**Originality:** 3
**Overall Recommendation:** 4
**Confidence:** 3

**Summary:**

The authors propose a novel and interesting foveated vision interface (FOVI) which is driven from human biology/neuroscience (human retina and primary visual cortex) to reformat variable-resolution sensor array to uniform and dense sensor manifold, with three contributions: (i) end-to-end kNN-convolutional architecture, (ii) foveated CNN method, (iii) LoRA based foveated adaptation of DINOv3 ViT model.

**Compliance With Llm Reviewing Policy:**

Affirmed.

**Final Justification:**

The authors have addressed most of my concerns within their rebuttal. As such, I raise my score to a 4.
The authors should see my raised suggestions for revision within the Rebuttal Acknowledgement.

**Key Questions For Authors:**

1.0 The authors have shown performance of different resource-constrained DINOv3 with FOVI variants. These have been compared using metrics: Accuracy 1 fixation, and Accuracy 3 fixations. Since the method is presented as a interface that can be applied across any vision architectures, the author should consider applying the FOVI interface to more than DINOv3 in Table 1 results. This would strengthen that FOVI is model agnostic beyond DINOv3 and CNN.

2.0 Since the method is presented as a interface, the authors should consider quantitative evaluation through cross dataset and task generalisation (COCO detection, ADE20K semantic segmentation) of their approach. This would strengthen the broader applicability of the authors foveated interface beyond classification.

3.0 The authors do not state the epoch length of their experiments. Please could the authors state the epochs trained. Additionally, the authors should justify the stopping criterion used within their training.

4.0 Including memory consumption statistics of the proposed foveated vision interface upon baseline would strengthen efficiency evaluation.

5.0 The authors may look strengthen their analysis by comparing against established efficiency methods (e.g., pruning or token reduction) to achieve a better understanding of the effectiveness of the proposed FOVI's efficiency.

**Limitations:**

Yes. However, an additional place to explore is that as the proposed FOVI modifies the spatial representation of the input image, it would be helpful to discuss how temporal (video) modelling may be affected by a foveation.

**Strengths And Weaknesses:**

Soundness:
Table 1. The authors provide 2 metrics for the effectiveness of their model being Accuracy 1 fixation, and Accuracy 3 fixations (further see 1.0 and 2.0)

Presentation:
This work is well structured and clearly written. One contribution (foveated CNN method) is not raised in the abstract.
Otherwise, no major structural or presentation issues to note.

Significance:
This work introduces a mechanism to reformat variable-resolution sensor array into uniform and dense sensor manifolds from neuroscience/biology inspiration from the human retina and primary visual cortex.
Driven by the efficiency of humans visual systems, this work is motivated to model foveation computationally via human neuroscience/biology.

Originality:
This work provides a novel insight of neuroscience/human biology (human retina and primary visual cortex) for their proposed foveated vision interface.
The authors proposed a new k-nearest-neighbour convolution technique, and the application of cortical magnification functions to CNN/ViT models.
The contributions are driven from the limitation within prior work of issues of anisotropic sampling (where sampling rate differs across polar angle and radius which results in warped receptive field shapes).

---

> ### Author Rebuttal · Authors · 2026-03-31
>
> # Response to F27X:
> We appreciate your positive comments and thoughtful questions.
>
> 1. Our approach is certainly architecture agnostic. For example, ViT-S+ and ViT-H+ are two different ViT variants of largely different sizes, and we showed competitive performance using both. We have also trained ViTs end-to-end with FOVI; we chose not to highlight those results only due to limited space. We would be glad to add these results to a camera-ready version, if you believe that would help to round out the paper’s claims of generality.
>
> 2. We appreciate the suggestions for further extensions of our work. We believe that segmentation and detection via foveation are interesting but non-trivial extensions of our work. We certainly believe it would be possible to make these extensions, and future work could address it. However, because our work is highly technically novel – introducing a new sensor model, a new kNN-convolution method for feature learning on the sensor manifold, and adaptations of two architecture classes to test the generality of our interface – we hope it is already of sufficient interest to the community.
>
> 3. Thank you for pointing this out. In most cases, we trained for 100 epochs, using a cosine-decay learning rate scheduler; the exception is ViT-H+ models, which were trained for 25 epochs as we found this was sufficient for them to reach saturation. We did not use any early stopping, and did not see any overfitting or reduction in validation loss over time that would have justified it.
>
> 4. This is a fair point which we can address in the camera ready. FOVI models have comparable memory usage with their non-FOVI matched-token counterparts, similar to latency and GFLOPs. Compared to the full resolution baseline, FOVI provides strong memory savings, particularly in training mode – where the computation graph and gradients must be stored, and thus can accumulate large patch-number-dependent memory usage. The memory savings are of strong interest to future applications in perceptive RL, where memory savings in a foveated vision model can be traded for scaling up other model components.
>
> 5. We appreciate the suggestion; however, as stated in our general response, our approach has the added benefit of reducing pixel count, which can be additionally useful beyond pure token reduction.
>
> Limitation: Indeed, due to the redundant content in videos, foveation is promising for video streams; this point will be added to discussion.
>
> # General reply to all reviewers
> ## Framing
> We believe the paper is best understood as introducing a biologically grounded sensing interface and processing mechanism for deep vision models, rather than as claiming state-of-the-art performance on every downstream benchmark or uniform superiority compared to other token reduction techniques. Concretely,  the submission establishes three concrete contributions: architectural generality, by demonstrating the interface in both CNN and ViT settings; a distinct technical advance over prior foveated approaches, through locally isotropic sampling together with kernel-mapped kNN-convolution that avoids the anisotropy of log-polar and warped-Cartesian methods; and a controlled empirical demonstration that reports not only accuracy, but also pixels, patches, GFLOPs, and latency. FOVI is intentionally a sensing interface, not only a token-reduction method. Broadly, we see this paper as establishing a core technique for scalable active vision, especially in settings where full-field high-resolution sensing is prohibitively expensive and sample count itself is a systems-level constraint.
>
> ## Correction:
> During rebuttal preparation, we identified an implementation error affecting the non-FOVI ViT baselines at non-standard (@64) resolution. After resizing the input and patch configuration, the RoPE module was inadvertently left with the original configuration, causing an incorrect determination of the number of visual vs. prefix (CLS and register) tokens. In practice, this affected Uniform @64 and Log-polar @64, while not affecting the standard non-FOVI @224 baselines or the FOVI / Weak-FOVI models. After fixing and re-running, the corrected results preserve the same overall ordering as originally reported, but with smaller deficits relative to FOVI: for (acc@1 fixation, acc@3 fixations) Log-polar@64 changes from (0.614, 0.664) to (0.643, 0.694), and Uniform@64 changes from (0.672, 0.711) to (0.693, 0.726). For reference, FOVI@64 is (0.700, 0.735) and Weak-FOVI@64 is (0.693, 0.725). This addresses Reviewer HDDR’s concern about the difference between Uniform @64 and Weak-FOVI @64, which is now essentially eliminated, while leaving unchanged the broader claim that FOVI provides competitive performance under strong sensing constraints and retains stronger performance than both the corrected matched-resource log-polar and uniform baselines.

---

> > ### Author Rebuttal · Reviewer_F27X · 2026-04-04
> >
> > I thank the authors here for their rebuttal. The authors have addressed most of my concerns here within their rebuttal. As such, I shall raise my score to a 4.
> > I suggest that the authors include 3. within their revised paper as this is critical for reproducibility of the methodology. I further suggest the authors include 1. in their revision.
> > Additionally, I agree with reviewer HDDR that this is an interesting direction and should be explored by the community.

---

> > > ### Author Response · Authors · 2026-04-08
> > >
> > > Thanks very much for your helpful review. We will certainly make the changes per the discussion. If possible, we'd greatly appreciate you officially changing the score to a 4 so it is fully reflected in the review system.

---

### Decision · Program_Chairs · 2026-04-30

**Decision:**

Accept (regular)

**Comment:**

FOVI presents a biologically grounded foveated interface that maps a retinal-like sensor array onto a dense, V1-like manifold and introduces a k-NN convolution that preserves local isotropy. The work demonstrates the interface on both CNNs (illustrating eccentricity-dependent receptive fields) and ViTs (showing competitive accuracy at reduced FLOPs). Reviewers F27X, GSBi, and tRtQ all rated acceptance after the rebuttal. Reviewer HDDR's concerns about limited evaluation and unclear efficiency trends remain, but he now endorses acceptance with revisions.

The study is confined to ImageNet classification and an AlexNet-style CNN, leaving performance on detection, segmentation, and larger modern backbones untested. Comparisons to token-reduction methods such as Swin, Pyramid, and larger-patch ViTs are absent. Addressing these gaps will strengthen the contribution.

Overall, the paper merits acceptance. It is recommended that the authors incorporate the reviewers' suggested revisions in the final version.